# Seismic control of large prehistoric rockslides in the Eastern Alps

Patrick Oswald [1✉], Michael Strasser [1], Christa Hammerl[2] & Jasper Moernaut [1]

Large prehistoric rockslides tend to occur within spatio-temporal clusters suggesting a common trigger such as earthquake shaking or enhanced wet periods. Yet, trigger assessment remains equivocal due to the lack of conclusive observational evidence. Here, we use high-resolution lacustrine paleoseismology to evaluate the relation between past seismicity and a spatio-temporal cluster of large prehistoric rockslides in the Eastern Alps. Temporal and spatial coincidence of paleoseismic evidence with multiple rockslides at ~4.1 and ~3.0 ka BP reveals that severe earthquakes (local magnitude $M_L$ 5.5–6.5; epicentral intensity $I_0$ VIII¼–X¾) have triggered these rockslides. A series of preceding severe earthquakes is likely to have progressively weakened these rock slopes towards critical state. These findings elucidate the role of seismicity in preparing and triggering large prehistoric rockslides in the European Alps, where rockslides and earthquakes typically occur in clusters. Such integration of multiple datasets in other formerly glaciated regions with low to moderate seismicity will improve our understanding of catastrophic rockslide drivers.

[1] University of Innsbruck, Department of Geology, Innsbruck, Austria. [2] Central Institute for Meteorology and Geodynamics, Vienna, Austria. ✉email: Patrick.Oswald@uibk.ac.at

Large rockslides are major landscape modifiers in mountainous regions[1] and often induce secondary and cascading hazards, such as rockslide-induced impulse waves[2] or valley damming with subsequent outburst floods[3]. Documented rockslides are mostly triggered by heavy precipitation[4,5] or severe earthquakes[6]. Some prehistoric rockslides were of extraordinary size, and if similar events were to occur today they would have devastating impacts. Our understanding of the causes of prehistoric rockslides is hampered by the lack of adequate observational data to test trigger hypotheses, and by the uncertainties related to rockslide modelling and to linking mechanic processes to mappable rockslide features[7]. As prehistoric rockslides show a tendency to occur in spatio-temporal clusters[8–10], a common trigger is often proposed, such as hydro-climatic change[9,11] or seismic activity[12]. Yet, instability of a large rock slope does not occur by a single strong disturbance, but rather forms the final stage of progressive rock slope weakening towards critical slope stability by a complex interplay of predisposition and preparation factors[13]. Besides lithological and structural control, predisposition in formerly glaciated mountainous regions is typically achieved by oversteepened topography and glacial debuttressing[13]. Progressive weakening of rock slopes acts on different time scales including long-term stress-release fracture propagation driven by deglacial unloading[14,15] (static fatigue), repeated earthquake-induced loading[7] (seismic fatigue) and seasonal pore pressure increase[16] (hydromechanical fatigue). In contrast to laboratory tests and in-situ rock slope monitoring of static and hydromechanical fatigue mechanisms[17], direct investigations of seismic fatigue are scarce[18] and difficult due to long reoccurrence time and the unpredictability of severe earthquakes, and therefore the role of seismicity in controlling large rockslides remains unclear.

In the Eastern European Alps, a spatio-temporal cluster of large prehistoric rockslides (4.4–3.0 ka BP) is documented for which the triggering mechanisms are unclear[19]. These rockslides originate in either massive carbonates (Fig. 1a; Northern Calcareous Alps) or in competent metamorphic bedrock (Austroalpine basement), involve large volumes ($25 \times 10^6$ m³ to $3.3 \times 10^9$ m³) and long run-out distances (up to 16 km), and have dammed major river valleys. Several severe historical earthquakes up to local magnitude ($M_L$) of 5.3 and epicentral intensity ($I_0$) of VII–VIII (EMS-98) attest that this region is one of the most seismically active areas in the Eastern Alps[20]. Seismotectonic activity in this intraplate setting is concentrated within the European plate[21] (Supplementary Fig. 1) and occurs at relatively shallow depth (mainly ~5–10 km). Therefore, heavy infrastructural damage ($I_0 = VIII$) can occur during earthquakes of only moderate magnitude. The recurrence patterns of $M_L > 5$ earthquakes are unknown due to the absence of paleoseismic data.

In mountainous regions, glacigenic lake basins can hold a high potential as long-term paleoseismic archives due to their accurately datable, high-resolution, and continuous sedimentary sequences since deglaciation (~17–18 ka BP) while being sensitive to record imprints of earthquakes[22]. The methodological principles of lacustrine paleoseismology have been well established in the last two decades by using e.g. multiple, coeval mass-transport deposits (MTD) overlain by a cogenetic turbidite in relatively deep and large lakes[23,24] or soft sediment deformation structures (SSDS) in small and shallow lakes[25,26].

In this work, we present the lacustrine paleoseismic records of the small, shallow lake Piburgersee and the larger, deeper lake Plansee (Supplementary Note 1) and their application to evaluate causal factors of large prehistoric rockslides. We infer a prominent role of seismic fatigue as preparatory as well as triggering factor of large rockslides based on temporal and spatial relationships between paleoseismic data and rockslide occurrence. Moreover, semi-quantitative paleo-earthquake data, such as

seismic intensity levels and minimum magnitude estimates, further constrain the seismic fatigue process.

## Results

**Piburgersee paleo-earthquake record.** The Holocene sediment succession of Piburgersee is composed of lacustrine mud with low background sedimentation rates (0.24 mm/a; Supplementary Figs. 2, 3). It is intercalated by different types of event beds (Supplementary Fig. 4), such as numerous flood deposits, three debrite intervals and a 2.5 m thick turbidite induced by a local rockfall impacting the lake floor at ~2.9 ka BP. At least one of the debrite intervals is related to the activity of the nearby Habichen rockslide (Fig. 1) at ~4.4 ka BP (Supplementary Fig. 4).

Furthermore, SSDS occur in eight distinct stratigraphic levels (SSDS 1-8; Fig. 2a; Supplementary Fig. 5; Supplementary Movie 1). Three-dimensional X-ray computed tomography data (CT) reveal (i) folds of intact (SSDS 3) or torn flood deposits (SSDS 7) within a ~10 cm thick bed of mixed lacustrine mud, (ii) an incipient breccia with isoclinally folded and sheared flood deposits overlying micro-faults (SSDS 2; Supplementary Discussion 1), and (iii) intraclast breccias with upward grading of tabular-to-round soft clasts of remnant lacustrine mud and flood deposits (SSDS 1, 4, 5, 6 and 8). CT structural analyses suggest a spectrum of increasing deformation from (i) – (iii): For SSDS 3, the intact folded flood deposit (Fig. 2b) indicates a lesser amount of shearing compared to SSDS 7, in which the flood deposit is slightly torn apart resulting in incipient occurrences of tabular intraclasts within an overall preserved stratification (Fig. 2c). For SSDS 2, progressive stratal disruption by tearing and shearing produced abundant tabular intraclasts and a barely preserved original stratification (Fig. 2d). The final stage of stratal disruption is represented by intraclast breccias with an upwards grading of clast-supported to matrix-supported texture. Random b-axis orientations of intraclasts corroborate in situ deformation as the underlying process (rose diagram in Fig. 2e; Supplementary Fig. 5), because gravitational flows would produce a dominant clast orientation[27].

Similar observations of SSDS in other lacustrine sedimentary successions have been validated to be induced by seismic shaking through correlation with severe historical earthquakes[25,26,28]. Furthermore, numerical modelling of seismically induced shear instability within stratified near-surface layers[29] reveals comparable progressive deformation structures as observed in our sedimentary data. A seismic trigger for SSDS in Piburgersee is further supported by the co-occurrence of micro-faults immediately below SSDS and by ruling out alternative aseismic causes for the observed SSDS (Fig. 2d; Supplementary Fig. 4). Therefore, we assign SSDS 1-8 to eight earthquakes (EQ-1 to EQ-8) that occurred between ~3.0 and ~9.9 ka BP (Supplementary Table 1, Supplementary Fig. 5).

Since Piburgersee has not recorded any of the historical earthquakes reaching seismic intensities up to VI (EMS-98) at the lake site, we infer the lake-specific intensity threshold for generating SSDS to be >VI (Supplementary Fig. 6). In analogy to quantitative constraints from earthquake-induced shear instability models[29], we consider that the progressive deformation sequence links to increasing seismic intensity above the intensity threshold >VI. Thus, we additionally define qualitative earthquake intensity levels for the eight paleo-earthquakes at Piburgersee (Fig. 2): EQ-3 and EQ-7 produced relatively lower, EQ-2 intermediate and EQ-1, −4, −5, −6, −8 higher intensities.

**Plansee paleo-earthquake record.** High-resolution reflection seismic data image the complete postglacial sedimentary infill of the central basin in Plansee, comprising a total of 64 MTDs that

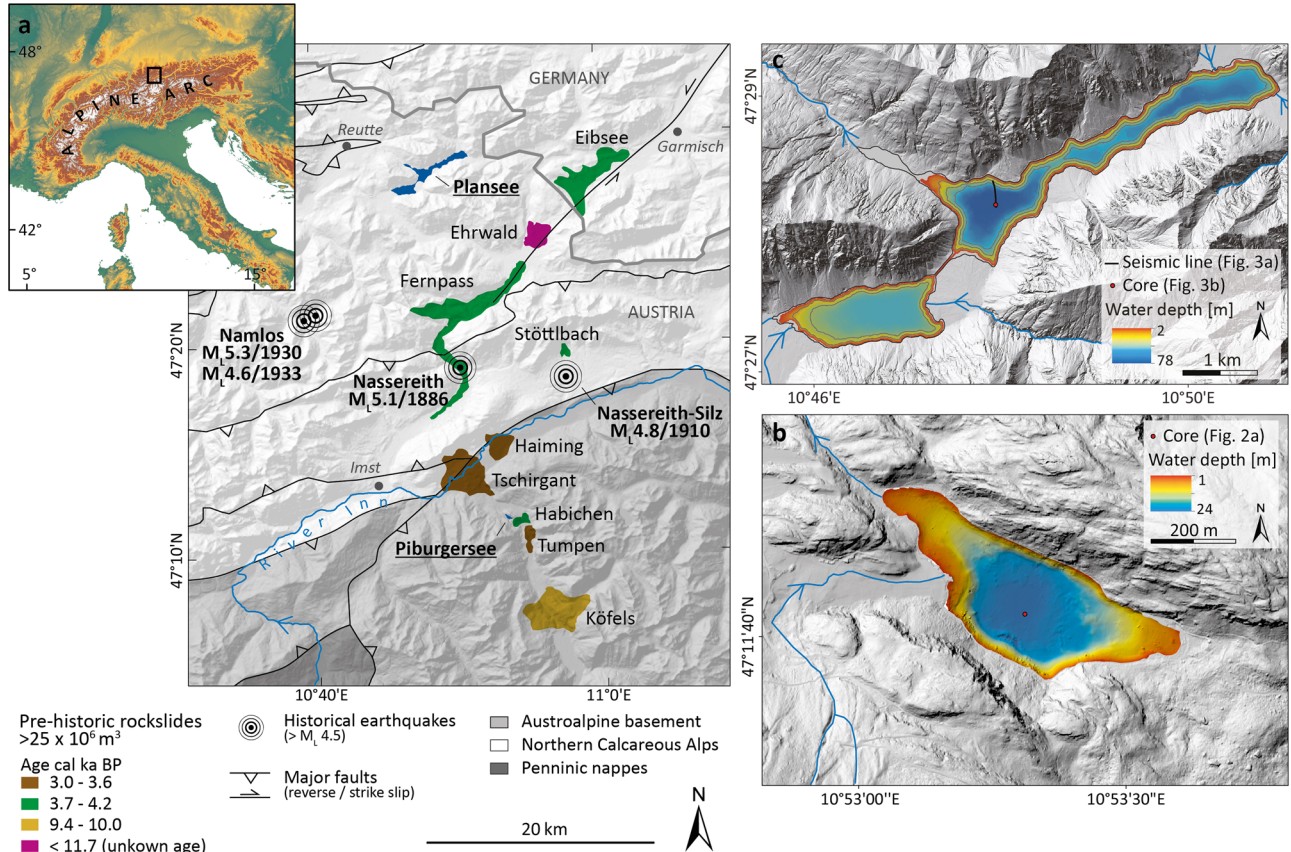

**Fig. 1 Large rockslides[19,35–38], historical severe earthquakes and the investigated lakes in the Eastern Alps. a** Study area comprises large rockslides (>25 × 106 m³) clustering in space and time affecting different geological units (Austroalpine basement, Northern Calcareous Alps), historical earthquakes $M_L$ 4.6–5.3 and location of the investigated lakes Piburgersee and Plansee. Major reverse and strike-slip faults[53] subdivide tectonic units or unit-internal nappes. Dashed line shows location of the Piburgersee-Plansee transect of Fig. 4. **b** Lake Piburgersee bathymetry and coring site of Fig. 2. Blue lines indicate the in- and outflow. **c** Lake Plansee bathymetry including seismic profile and coring location of Fig. 3 in the central basin. Contour distance is 20 m. Blue lines with arrows indicate main in- and outflows. Onshore digital elevation models are derived from Copernicus Land Monitoring Services and Land Tirol—data.tirol.gv.at.

relate to eleven distinct seismic-stratigraphic event horizons, each of which contain 2 to 12 MTDs (events A-K in Fig. 3a and Supplementary Figs. 7, 8 and 9). Multiple coeval MTDs corresponding to a stratigraphic event horizon provide strong evidence for simultaneous failure of several subaquatic slopes[23]. Further indication for the synchronicity of subaquatic slope failures is provided by the occurrence of a megaturbidite with ponding geometries and low-amplitude facies, immediately overlying the multiple MTDs of events C and E[30] (arrows in Fig. 3a).

In the sediment core, the Holocene succession is composed of thin-laminated clayey silts with intercalated <2 cm thick flood deposits. Thick laminated to thin bedded clays build up the Late-Glacial sedimentary succession (Fig. 3b; Supplementary Figs. 10, 11). Holocene seismic-stratigraphic event horizons A to E (Fig. 3a) correlate to 5–35 cm thick event beds, each composed of an amalgamated turbidite (Fig. 3b) with a thin clayey-silt top (Supplementary Fig. 12; Supplementary Movie 2). This indicates deposition from individual turbidity currents at the coring site within seconds to minutes before settling of suspended fine-grained clayey-silt, and thus synchronicity of the corresponding slope failures[31].

Such synchronous subaquatic slope failures require an external trigger, which we assign to the occurrence of five earthquakes that struck the Plansee region over the last 10,000 years (EQ-A to EQ-E; Supplementary Table 2). This interpretation is analogous to many studies that have established a paleo-earthquake proxy

based on multiple coeval MTDs and correlated amalgamated turbidites in lakes[22,24,32]. Moreover, event A correlates to the Common Era (CE) 1930 Namlos earthquake ($M_L$ 5.3; Supplementary Fig. 13). Positive and negative sedimentary evidence of historical earthquakes up to $M_L$ 5.3 defines the lake-specific earthquake-recording threshold to seismic intensity ≥VI (Supplementary Fig. 6).

Additional qualitative constraints on seismic intensity are inferred from the occurrence of enhanced clastic sediment input in the aftermath of prehistoric events C and E (Fig. 3b; Supplementary Fig. 12; Supplementary Movie 2). We interpret this as postseismic sediment flux resulting from landscape response to relatively stronger earthquakes causing terrestrial landslides in the lake catchment[26,33]. Such postseismic landscape response has been calibrated to at least 1–2 intensity levels higher than what is required to trigger subaquatic failures[33]. Thus, we interpret earthquakes EQ-C and EQ-E to have reached higher intensities than EQ-A, -B, and -D, well above the threshold intensity ≥VI at the lake site (Fig. 3). This scaling is further supported when considering a possible turbidite thickness–intensity relationship[34], because the stronger earthquakes EQ-C and EQ-E correspond to the thickest turbidites in the record (35 and 15 cm, respectively; Supplementary Fig. 12). Following this concept, EQ-B (10 cm) has produced relatively intermediate intensity and EQ-A and EQ-D (~5 cm) lower intensities at Plansee.

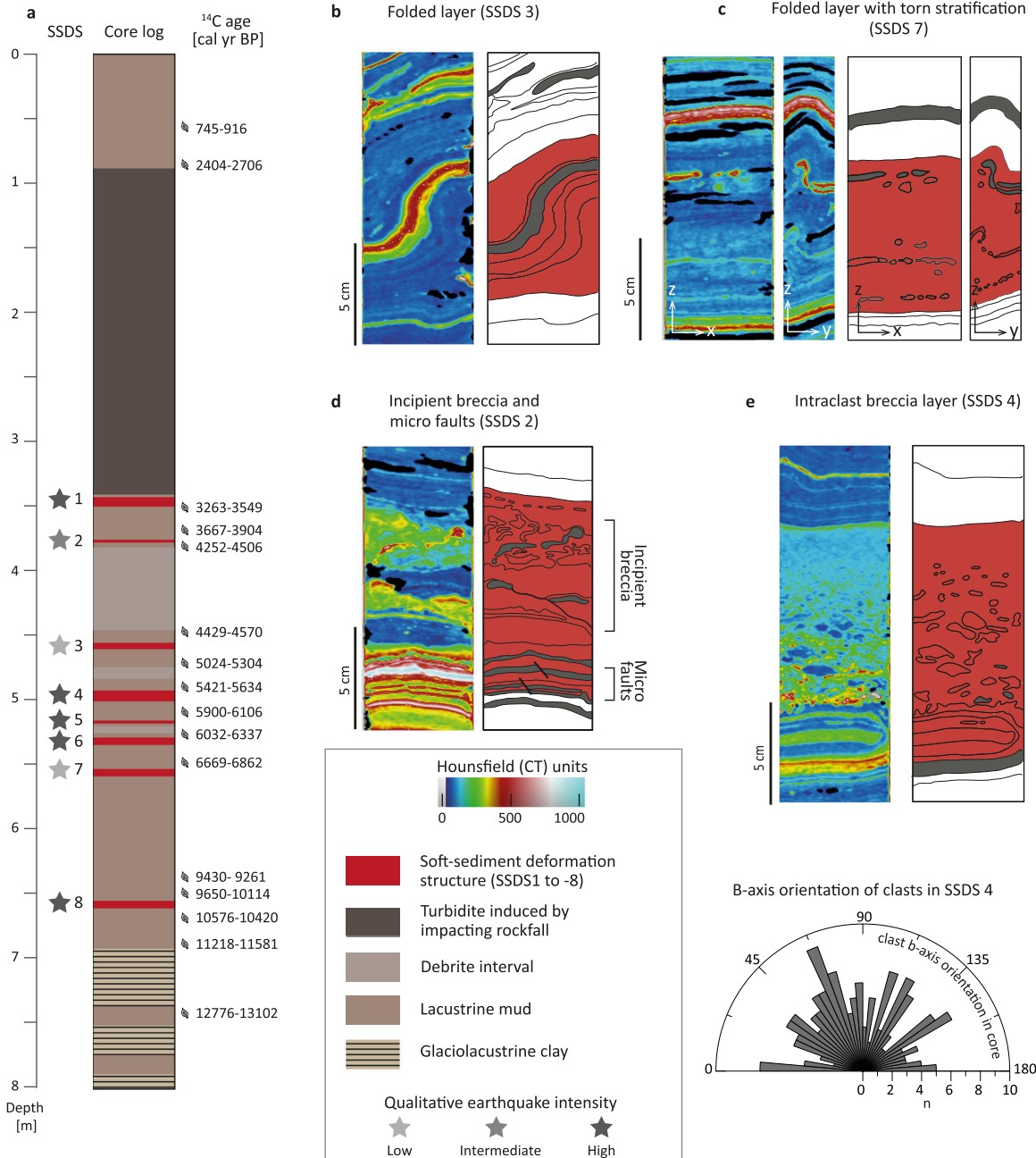

**Fig. 2 Earthquake-induced soft sediment deformation structures (SSDS) in Piburgersee. a** Core lithology with event deposits and calibrated 95% age-range of [14]C samples. Interpretation of the low- to high qualitative earthquake intensities (grey-scaled stars) is based on the observed succession of increasing deformation from SSDS (**b**) to (**e**) explained in main text. **b–e** CT images supported by schematic sketches of SSDS (red). These are under- and overlain by undeformed sediment (white). High CT-density layers are coloured grey. Rose diagram displays randomly orientated *b*-axis of clasts within the intraclast breccia layer SSDS 4.

**Spatio-temporal relation between earthquakes and rockslides.** Our high-resolution paleoseismic data document an enhanced seismicity period between ~7.0 to ~3.0 ka BP in the southern area, and a more evenly distributed earthquake recurrence pattern in the northern area (Fig. 4). Seven of the ten large Holocene rockslides occurred between ~4.4 and ~3.0 ka BP[19,35–38] (Supplementary Table 3) at the end of the enhanced seismicity period (Fig. 4). Strikingly, the large rockslides Eibsee and Fernpass, as well as Tschirgant and Haiming, coincide within the overlap-age ranges of the paleo-earthquakes recorded in both lakes at 4.1 ± 0.1 and 3.0 ± 0.2 ka BP, respectively (Fig. 4; Supplementary Figs. 3, 11, 14). The possibility of earthquake imprints generated by

rockslide-induced shaking is very low, because i) the largest rockslide (Köfels, 3.3 km[3]) did not induce a SSDS in Piburgersee despite its short distance (9 km); ii) empirical relations between rockslide-induced shaking and rockslide volume for the Eastern Alps[39] suggest only a $M_L$ 2.5-3.5 for the rockslides in our study area, insufficient to generate seismic intensities above the thresholds of >VI and ≥VI. Thus, our lake paleoseismology dataset provides independent observational support for the earthquake-triggering hypothesis for these large rock slope failures. An alternative trigger mechanism formed by extreme precipitation events or extended wet periods in the study area can be reasonably ruled out, because no clear coincidence of enhanced

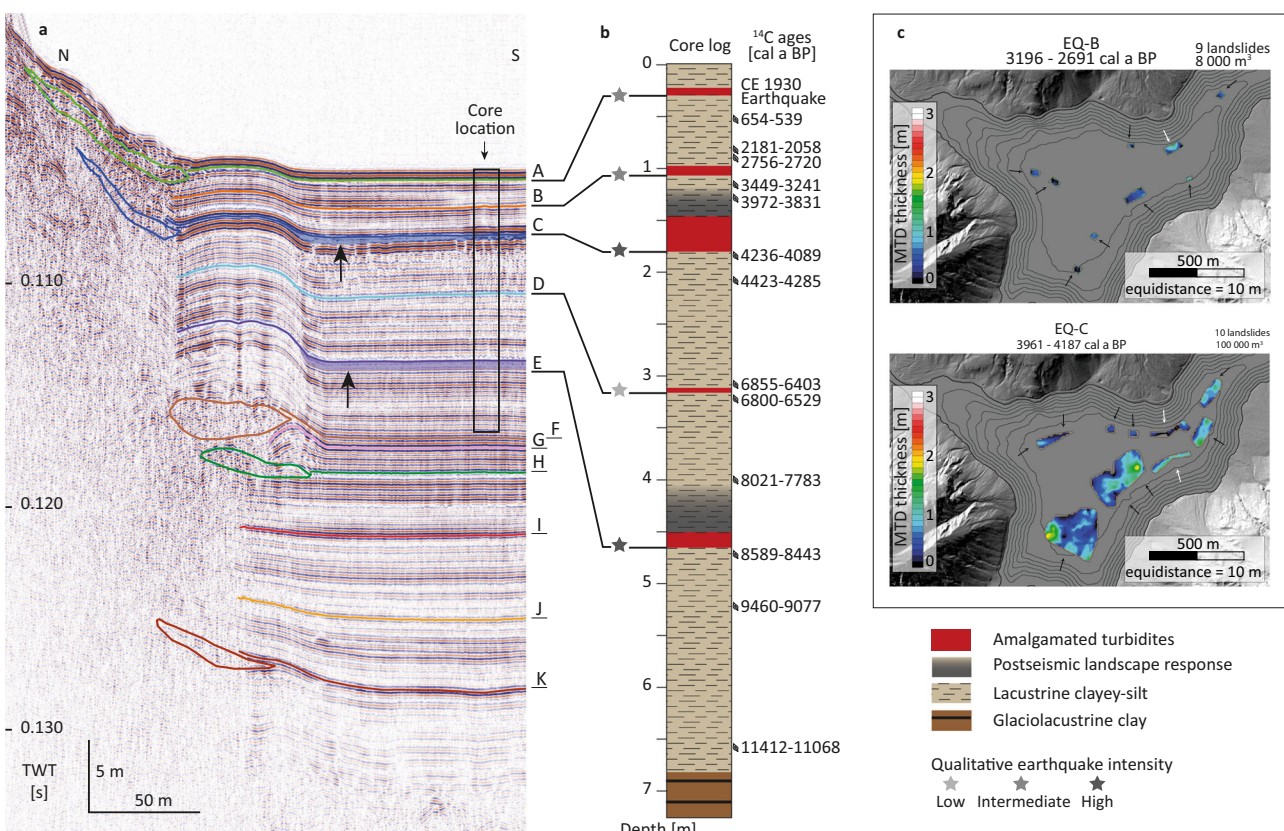

**Fig. 3 Earthquake-induced multiple, coeval MTDs in Plansee. a** MTDs and event horizons A to K on a seismic profile of Plansee. Black arrows indicate transparent megaturbidites with ponding geometries. Locations of seismic profile and core are shown in Fig. 1c. MTDs on seismic profiles correlate to amalgamated turbidites (red in **b**) in the sediment core. **b** Core lithology with amalgamated turbidites (red) and associated postseismic landscape response (grey) and 95% age range of calibrated $^{14}$C samples. Interpretation of qualitative earthquake intensities (grey-scaled stars) are based on postseismic landscape response[33] and relative turbidite thickness[34] explained in the main text. **c** MTD distribution and thickness maps for earthquakes EQ-B and EQ-C (Supplementary Table 2) demonstrate coeval failure of multiple hemipelagic (black arrow) and deltaic (white arrow) slopes (see Supplementary Figs. 7, 8). MTD maps of all event horizons are presented in Supplementary Fig. 9.

precipitation with the ~4.1 and ~3.0 ka BP rockslides can be observed (Supplementary Discussion 2, Supplementary Fig. 15).

For each of the 4.1 ± 0.1 and 3.0 ± 0.2 ka BP episodes a single-earthquake scenario is very plausible, but we cannot fully exclude a multiple-earthquake scenario covering maximum 100–200 years with sufficiently spaced epicentres so that each lake contains a single imprint (Supplementary Discussion 3). However, we observe a spatial correlation between the largest earthquake imprints in the lakes and the rockslide locations (Figs. 1, 4; Supplementary Table 3): the northern Fernpass and Eibsee rockslides at ~4.1 ka BP coincide with a large imprint in the northern Plansee and intermediate imprint in the southern Piburgersee, whereas the southern rockslides Tschirgant and Haiming at ~3.0 ka BP relate to an intermediate imprint in Plansee and large imprint in Piburgersee. This spatial relationship further corroborates the interpretation of seismically-triggered large rockslides and supports the single-earthquake scenario for the ~4.1 and ~3.0 ka BP episodes with epicentral locations more towards the northern and southern part of the study area, respectively.

Following a single-earthquake scenario that simultaneously induces seismic intensity >VI at both Plansee and Piburgersee (30 km apart), such an earthquake would have had a minimum magnitude of $M_L$ 5.5 (Supplementary Method 1). This minimum $M_L$ 5.5 is slightly higher than the CE 1930 $M_L$ 5.3 earthquake, which only caused a small sedimentary imprint in Plansee (Supplementary Fig. 6). An upper magnitude bound of $M_L$ 6.5 in

the study area is obtained from focal-depth distributions inferred from macroseismic data[20] and is supported by a worldwide relationship between maximum magnitude and plate convergence rate[40]. Earthquakes in this estimated magnitude range ($M_L$ 5.5–6.5) are larger than the historically-documented events in the study area ($M_L$ up to 5.3) and produce epicentral intensities of VIII¼–X¾[20]. According to empirical earthquake magnitude-to-rockslide volume relations (ESI-2007)[41], such intensities are capable of triggering the documented rockslides with volumes of ~$10^7$–$10^9$ m³. In conclusion, such quantitative considerations, along with the spatial and temporal correlation of earthquake imprints and rockslides, attest that the large rockslides were triggered by severe seismic shaking.

**Seismic fatigue and instability of rock slopes.** Our paleoseismic data document initiation of an enhanced seismicity period in the internal Eastern Alps at ~7.0 ka BP before the spatio-temporal rockslide cluster (~4.4–3.0 ka BP; Fig. 4). At least five severe earthquakes hit the southern study area (Fig. 4; EQ-3 to EQ-7) without evidently triggering any large rockslides before the Tschirgant and Haiming rockslides took place at ~3.0 ka BP. Similarly, the Eibsee and Fernpass rockslides at ~4.1 ka BP are preceded by at least two (EQ-D, EQ-E; Fig. 4) but potentially up to eight earthquakes since deglaciation of the Plansee basin (Supplementary Fig. 9). These observations imply that earthquakes are more important for preparing rock slopes towards

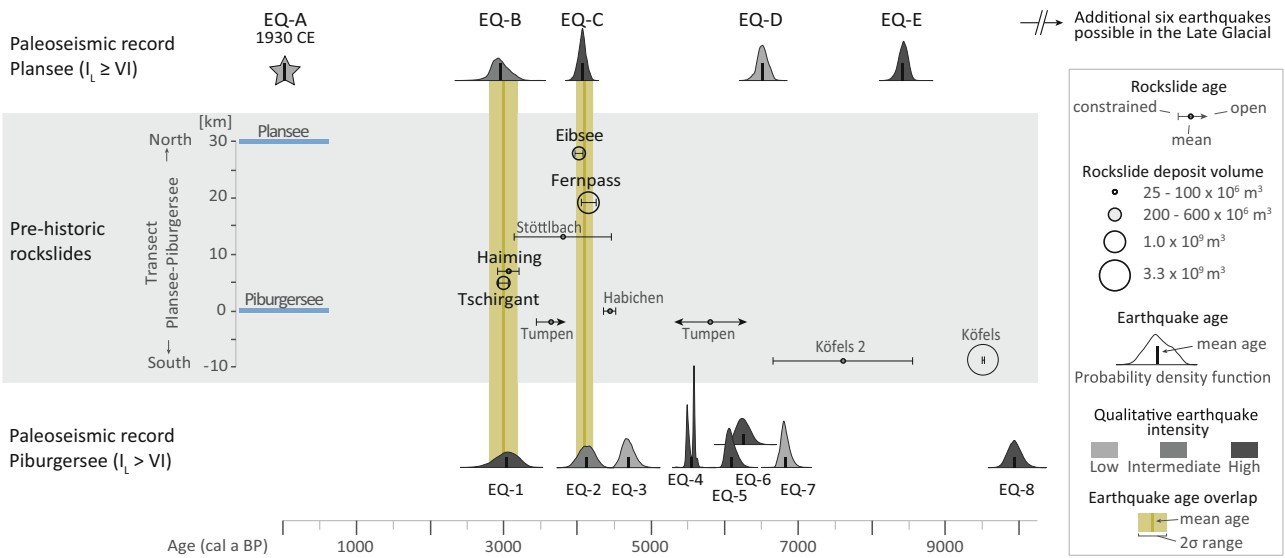

**Fig. 4 Comparison of the paleoseismic records from Plansee (top) and Piburgersee (bottom) with previously documented prehistoric rockslides**[19,35-38]. Significant age overlap of earthquakes recorded in both lacustrine archives at 4.1 ± 0.1 and 3.0 ± 0.2 ka BP (Supplementary Fig. 14) coincide each with at least two rockslides. Earthquakes are displayed as probability density functions derived from Bayesian age-depth modelling. Recorded (paleo-)earthquakes reached local intensities ($I_L$) of ≥VI and >VI at Plansee and Piburgersee, respectively (see Supplementary Fig. 6 for intensity threshold calibration) and are further classified into qualitative earthquake intensity levels based on sedimentological criteria (see main text). Ages of compiled prehistoric rockslides are displayed as constrained or open age ranges dependent on the applied age-dating methods (Supplementary Table 3). Deposit volume of rockslides is shown by dimension of circles. All rockslides occur within ~40 km distance and are projected perpendicular to a hypothetical north-south transect from Plansee to Piburgersee defining the y-axis (transect location in Fig. 1a).

failure[7,18] (seismic fatigue) than for being the ultimate trigger: Repeated severe seismic shaking causes incrementally increasing damage in a rock slope[7]. This, together with long-term static[14,15] and hydromechanical fatigue[16], progressively weakens the rock slope towards a critical state. Once critical state is reached, a minor disturbance, such as another earthquake or heavy precipitation may initiate the final acceleration of the slope. For instance, the giant Köfels rockslide at ~9.5 ka BP[35], as the second largest rockslide in the entire Alps, is not earthquake-triggered according to our paleoseismic data, but is shortly preceded by at least one severe earthquake (EQ-8, ~9.9 ka BP, Supplementary Table 1). This earthquake (EQ-8) falls into a period of Alpine-wide enhanced seismicity between 9.5 – 9.9 ka BP during which some of the largest rockslides in the Alps took place (e.g. Flims, Köfels)[42].

Mid-Holocene rockslide clusters occur at several locations in the Alps and have been considered to be either related to a phase of enhanced seismicity[12] or hydro-climatic change[9,11]. Hydromechanical fatigue can be considered as an important long-term preparatory factor for the rockslides in the study area, especially relevant during wet periods (Supplementary Fig. 15). However, no correlation can be observed between rockslide activity and enhanced precipitation periods inferred from available hydro-climate records near the study region (Supplementary Discussion 2; Supplementary Fig. 15). This suggests a subordinate role for hydro-climatic change in triggering these rockslides. Altogether, this study provides the first conclusive observational evidence that severe seismic shaking initiated the final acceleration phase of several prehistoric large rock slope failures in the Eastern Alps. Moreover, it supports the general concept of progressive seismic fatigue as key preparatory factor for large rockslides in comparable, formerly glaciated mountain regions with low to moderate seismicity. Such regions are typically characterised by episodic, clustered and migrating paleoseismicity[43–45] with millennial-scale recurrence rates. We propose that regional episodes of seismic fatigue explain why large prehistoric

rockslides often occur as spatio-temporal clusters, such as documented for the intraplate European Alpine region[9,11]. High-quality, multi-site paleoseismic records are required to document and better understand enhanced seismicity periods, to evaluate the role of seismic fatigue for other prehistoric rockslides and to constrain their ultimate trigger.

## Methods

**Reflection seismic data**. On both lakes, reflection seismic data were acquired using a single-channel 3.5 kHz Geopulse pinger resulting in a theoretical vertical resolution of ~10 cm. A bandpass filter (2.5–6.0 kHz) was applied and seismic interpretation was done in IHS Markit Kingdom Suite 2018. In Plansee, MTDs and their equivalent seismic-stratigraphic horizons were mapped in the central basin in order to recognise synchronous MTD events (Supplementary Fig. 7), characteristic for earthquake triggering[22]. Thickness grids of MTDs (Supplementary Fig. 9) were calculated with Surfer 10 using simple kriging interpolation and assuming an acoustic velocity of 1500 ms⁻¹.

**Bathymetry data**. High-resolution bathymetry data of Plansee (Fig. 1c) were acquired by a Kongsberg EM2040 echo sounder (University of Bern) operating on 300 kHz in October 2019. For positioning, a Leica GX1230 + GNSS receiver was used in combination with real-time kinematic corrections (RTK) provided by EPOSA. At Piburgersee (Fig. 1b), a Kongsberg GeoSwath plus echo sounder operating at 500 kHz together with a Trimble SPS855 GNSS receiver was used. The resulting point clouds were rasterized resulting in a bathymetric map with 1 m horizontal and a few decimetres vertical resolution.

**Sediment core analyses**. In 2018, sediment cores were recovered by an UWITEC percussion piston coring system (ETH Zürich) in order to date the sedimentary sequences (Supplementary Tables 4, 5; Supplementary Figs. 3, 11), evaluate the seismic interpretation in Plansee (Supplementary Figs. 7, 9) and investigate SSDS in Piburgersee (Supplementary Fig. 5). Sediment cores were scanned for X-ray computed tomography using a Siemens SOMATOM Definition AS at the Medical University of Innsbruck with a voxel size of 0.2 × 0.2 × 0.3 mm³. For CT data visualisation and b-axis measurements in SSDS of Piburgersee, the software VGstudio (v3.3) was used. Bulk density, p-wave velocity (at 0.5 cm resolution) and magnetic susceptibility (at 0.2 cm resolution) were measured by a GEOTEK Multi-Sensor Core Logger (MSCL) at the Austrian Core Facility of the University of Innsbruck (ACFI). Split cores were imaged using a Smartcube Camera Image Scanner (ACFI) and colour variability of the core images were enhanced using histogram equalisation. For the composite core of Piburgersee, elemental variation

measurements were analysed by the ACFI Itrax-XRF Core Scanner using a Mo X-ray tube with 30 kV, 35 mA on 0.1 cm resolution with 5 s exposure time.

**Event dating**. AMS $^{14}$C analyses on terrestrial leaf and needle macro-remains were performed at the Ion Beam Physics Laboratory of ETH Zürich and calibrated using the IntCal13 calibration curve. Short-lived radionuclide ($^{210}$Pb, $^{226}$Ra and $^{137}$Cs) activities were measured at EAWAG (Dübendorf, Switzerland) using CANBERRA and Princeton 146 Gamma-Tech germanium well detectors. Constant flux—constant sedimentation rate (CFCS) modelling on xs$^{210}$Pb activities of the uppermost sediments of Plansee was performed with the SERAC software package in R[46]. Paleoseismic event ages are based on Bayesian age-depth modelling with the Bacon v2.4 software package in R[47]. For the age-depth models, all macroscopically visible event beds (>5 mm) were extracted from the sediment depth (Supplementary Figs. 3, 11).

**Earthquake data**. Earthquake magnitudes are given in local magnitude ($M_L$) and seismic intensities correspond to the European macroseismic intensity scale (EMS-98). The calibration of the lake-specific intensity threshold of recording earthquakes is based on intensity data points derived from contemporary sources for two severe historical earthquakes[48] (Supplementary Fig. 6, Supplementary Tables 6, 7). The minimum magnitude calculation (Supplementary Method 1) is based on the latest intensity-prediction equation[49] and conversion between moment magnitude and local magnitude[50] based on earthquake data from the Austrian Alps.

## Data availability

The lacustrine geophysical and core datasets of Plansee are available on zenodo at https://doi.org/10.5281/zenodo.4382341[51]. The lacustrine core datasets of Piburgersee are available on zenodo at https://doi.org/10.5281/zenodo.4382482[52]. The Austrian earthquake catalogue is available from the ZAMG (Central Institute for Meteorology and Geodynamics, Austria) upon reasonable request.

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

## Acknowledgements

This research is supported by the European Regional Development Fund Interreg V-A Italy-Austria 2014-2020 (project ITAT3016-Armonia), the Tyrolean Science Fund (TWF): UNI-0404-2151, the Austrian Science Fund (FWF): P30285-N34 and the Austrian Academy of Sciences ÖAW (ESS-IGCP-project S4LIDE-Austria). We thank M. Erhardt, G. Degenhart and W. Recheis for the medical CT measurements at the Medical University of Innsbruck, J.-J. S. Huang for XRF-scanning, S. Fabbri and M. Aufleger for the bathymetry acquisitions, and all members of the coring crews on Piburgersee and Plansee. Furthermore, we thank C. Spötl and C. Prager for scientific discussions and P. Tropper for petrological analyses. Further acknowledged are I. Hajdas for radiocarbon analyses and N. Dubois for short-lived radionuclide dating. Land Tirol—data.tirol.gv.at is thanked for providing the DEM data. IHS Markit is acknowledged for their educational grant program providing the Kingdom seismic interpretation software.

## Author contributions

J.M., M.S. and P.O. designed the study. P.O., J.M. and M. S. acquired lacustrine data and interpreted results. C.H. acquired historical earthquake data. P.O. wrote the manuscript and produced the figures with input from all co-authors.

## Competing interests

The authors declare no competing interests.
