## [Peer Review File · Nature Communications]

REVIEWER COMMENTS

Reviewer #1 (Remarks to the Author):

The manuscript presents support for the seismic priming and triggering of clusters of large landslides in the eastern European Alps. This support comes from the coincidence in timing of landslide clusters with evidence from two lakes of seismic disturbance of lake sediments and pulses of sediment entering the lakes. The data presented in the manuscript focuses on detailed descriptions of single sediment cores at each lake and seismic profiling provided at one of the lakes. The idea of a seismic trigger for the landslides, and therefore large prehistoric earthquakes in the Alps is not new, but this study provides strong support for this by combining two independent datasets – lacustrine and terrestrial evidence. After having identified events in both lakes that overlap in timing with dated landslide clusters in the surrounding mountains, the authors attempt to evaluate the thresholds of seismic shaking needed to generate seismic disturbances to the lakes, and assess the likely magnitude and epicentral area of the earthquakes. They also identify earthquakes in their lake records for which there is no current evidence of landsliding, and postulate that seismic priming may be responsible for bringing slopes to a point of failure in subsequent earthquakes. The data presented appear to be robust and well explained with well-constructed figures and supporting information. The manuscript is a welcomed addition to the debate on the cause of landslide clusters in the Alps and the possibility of high-magnitude earthquakes affecting this region in the recent geological past, and will hopefully promote similar work in other localities. Below are some suggestions for how this manuscript might be strengthened:

1. Given that the findings strongly suggest large magnitude earthquakes affecting the region in the recent geological past, there is perhaps a few aspects of this that could be elaborated on further by the authors. What is the societal or hazard implication of this finding – does it suggest the likely possibility of similar-sized earthquakes with the generation of large landslides in populated parts of the Alps? This is perhaps implied but it could be communicated more explicitly. Are there any existing investigations of Holocene fault surface ruptures that corroborate evidence of earthquakes in the region at the approximate dates suggested, or is the lack of surface rupturing fault traces the reason for why lake archives and landslides are needed for investigating the paleoseismicity in this region?

2. The idea of seismic priming of hillslopes and the evidence for this could be tightened up or discussed further, as this is perhaps the weakest part of the manuscript.

- a. It is possible that the landscape was more sensitive to earthquake-triggering in the Mid-Holocene when groundwater levels were more elevated by the climate at that time. It might be useful to discuss this point with specific reference to the papers that support a climatic link to the temporal clustering of landslides – e.g. Zerathe et al. 2014 which you cite or Panek 2019 (<https://doi.org/10.1016/j.earscirev.2019.05.015>) for a wider review of the topic. If such a climate influence is to be accepted then an alternative hypothesis for the reason why earthquakes did not trigger slope failures in the preceding earthquakes is that climatic conditions were more favourable for stability in those times, rather than (or in addition to) seismic priming.

- b. There are other studies that have investigated seismic priming that might be worth at least brief reference to. For example Parker et al. 2015 (doi:10.5194/esurf-3-501-2015) who provide support for such a process, and Brain et al., 2017 who challenge the notion that precursor seismic events always lead to weakening and priming (<https://doi.org/10.1002/2016GL071879>).

- c. An idea for explaining the timing of paraglacial landslides long after glaciers have retreated is the stress release fracturing (static fatigue) that rock slopes may experience. For example, stress release has been postulated to have played a role in priming earthquake-triggered failures in the Scottish Highlands (Ballantyne et al., 2014; <https://doi.org/10.1016/j.quascirev.2013.12.021>). Of course, the landslide clusters in the Alps may have been a result of a combination of static fatigue and cyclic (seismic) fatigue, and both are worth considering.

- d. Others have pointed to deep seated gravitational slope deformation (i.e gradual and potentially

aseismic deformation leading to failure) influencing some of the landslide clusters in the Alps (e.g. Ostermann and Sanders 2017; <https://doi.org/10.1016/j.geomorph.2016.12.018>). It may be worth commenting on this.

3. The attempt to define earthquake intensity levels from the occurrence of SSDS or other sediment events is interesting and overall such an approach seems reasonable. However, I suspect that the earthquake characteristics (depth, frequency spectra, wave propagation directivity, duration) and the wave interactions with topography (e.g. amplification or de-amplification, which in turn are dependent on wave characteristics) might influence sediment response in a lake, in addition to earthquake magnitude and shaking intensity. This could perhaps help explain the absence of sediment disturbance evidence at the time of the Köfels landslide. Perhaps some comment on such effects would be worth adding?

Minor comments on clarity:

Main document:

5. Line 16 of abstract: consider omitting the word 'exceptionally'. ML >5.5 is probably not exceptionally strong...though I do live on an active plate boundary so perhaps I have a different tolerance ;-). Consider also omitting the word 'ultimately' on the same line.

6. Line 17 of abstract: perhaps add the words 'are likely to have' to this sentence to help convey the uncertain nature of this statement.

7. Line 20 of abstract: it is unclear what is 'new' about the methods presented. Again on L67-68 you imply that this research is novel in using lake sediments to evaluate the causes (i.e. earthquake triggers) of landslides. This is not really true. Perhaps what is novel is the integration of these two datasets in the eastern Alps.

Supplementary Information document:

8. Line 145-147. This sentence is unclear. Are you saying that the geochemistry of the SSDS is similar to the underlying undeformed sediments? Also if this is the intent, it is hard to see why this is considered evidence for the deformation to have occurred at the sediment-water interface. If the deformation occurred below the sediment-water interface it presumably would also share the same geochemistry as any adjacent undeformed sediments.

9. L152-155. This sentence is a bit hard to follow. Consider revising.

10. L215. Should be 'extent' not 'extend'

11. Supplementary Figure 9: What are the red triangles on the map of Event Horizon G?

Yours sincerely,
S McColl

Reviewer #2 (Remarks to the Author):

I really enjoyed reading the manuscript. This is a nice piece of multidisciplinary research in geosciences bringing together, lake sedimentology, landslides and seismicity. The figures are clear, the text is very well readable. However, there are several remarks and amendments marked in the annotated manuscript attached.

Starting with the most critical issue: the earthquakes and magnitudes.

The 1930/10/08 eq was deep, up to 35 km are estimated and had a thrust mechanism. Those deep eqs do generally not produce surface ruptures, as they are "blind" to the surface. You do not mention the date, and that there was heavy rainfall (after, but unknown before). So, if there are no surface evidence, and little deformation described (some rockfalls, but no major severe

landscape changes), how can you find the "postseismic landscape changes" in your archives? I have the feeling this is over-interpretation, also you do not provide alternative thoughts? Also, please explain a 2 m thick "turbidite" in the archive, though it is really a small archive? Where does the sediment come from? Explain better, as the other layers are max. 35 cm thick. Even megaturbidites in deep water hardly form that thick layers.

- secondary is not cascading!

- climatic deterioration - evidence and what does this mean? It is rather subjective, I would say: more rain, colder, more snow, less precipitation.....

The distance or proximity to active faults (and not only epi/hypocentres) is important, I did not find that? Are these faults capable for the eqs needed? Why put it in the center of the "transect"? Makes no sense to me, and is just a game to play.

The area is really not the "most seismic area in the E. Alps), by far not. No M 7 is ever found in the Alps (maybe in Vienna Basin and Friuli).

You never discussed, because you relate 1 SSDS to 1 eq, but is this true?

Can a large eq overprint older structures? Can an eq cause SSDS in several of your layers? Why not? I can see huge differences in the deformation style, e.g semi-brittle deformation and even microfaults? So, this layer was "harder" more rigid to deformation? Discuss this please.

And this also effects your age discussion and clustering. Discuss this as well.

The next problem are sedimentation rates, between the 1930 and the present deposits are c. 25 cm in 90 years, but the date before is c. 500 BP and sediment accumulated also 25 cm? Explain variation in rates.

You are citing much larger eqs for the "postseismic sediment flux" and landscape response, like the Alpine fault (M 7.6, which means c. 1000 times more energy was released; the Turkish ones are min. 5.5, but slightly higher up to 6.5) - is this a wishful thinking? Re-check and provide (much) better arguments.

More in the annotated ms.

The methods (geophysical and sedimentological, dating etc.) are excellent.

Abstract:

I can not see any clustering in the eq data? You claim for $M > 5.5$ but there is no evidence, neither historical nor instrumental provided. You speak of series of "strong" eqs. For my feeling strong is larger than M 6.5. Your eqs are moderate, think of the energy released. And also think of the focal depth. You never discussed peak acceleration values? Why those are important for triggering landslides, but again it is a function of depth. I guess your eqs are around 0.4-0.5 g, so far less in comparison to what has been observed in Norcia or Finale Emilia (blind as well).

Also, I have problems of the "weakened" rock masses prone to fail...(because of repeated eqs). What about changing weathering rates or post-LGM (deglaciation debris?).

Videos: add scale.

The manuscript needs major revisions, as too many evidences and arguments are around the threshold value during eq shaking. And some interpretations are not sound.

But as the ms is interesting and novel, the ms should be elaborated better.

regards

Klaus Reicherter, RWTH Aachen University

Reviewer #3 (Remarks to the Author):

To the editor and authors,

The manuscript title 'Seismic control of large prehistoric rockslides in the Eastern Alps' submitted to Nature Communications by Oswald et al. uses lacustrine paleoseismology to explore the link between seismicity and the occurrence of large rockslides in the European Alps. The authors present lacustrine paleoseismic reconstructions of unusually high quality. More importantly, they apply them in a novel way to test long standing hypotheses about seismicity as both a preparatory and a triggering factor in the initiation of large rockslide clusters. The authors use their impressive lacustrine paleoseismology dataset to argue that seismic fatigue is the main preparatory factor causing clustering of large rockslide occurrence in the European Alps. The work is suitable for publication in Nature Communications for three main reasons: i) the work presents very high quality lacustrine paleoseismology datasets; ii) these high quality datasets are applied in a very novel way to address a scientific question that has remained poorly tested until now; and iii) both the results and approach have global applicability because they elucidate the processes that control large rockslide initiation and hence have relevance for landscape evolution and hazard in mountain landscapes.

The manuscript is very well written and the authors have done an excellent job of compiling a detailed supplement for the specialist reader. It is my opinion that this is the perfect set up for a Nature Communications paper. Generally, all the critical information for a general reader to understand the arguments developed in manuscript is present in the main text, while the supplement provides the detail required by the specialist reader who wants to mine deeper in to the dataset that supports the author's conclusion.

While I commend the authors for their attention to detail and ingenuity, I have one major concern that must be addressed before the manuscript is accepted for publication. Therefore, I recommend minor/moderate revisions before the manuscript is accepted for publication. I have also identified a series of minor issues that are outlined below, referenced by line number.

Kind regards,

Anon.

Major concerns:

Climatic change as a confounding variable in the author's natural experiment.

In their introduction the authors acknowledge that both seismicity and climatic change are often invoked to explain temporal clustering of large rockslides (line 33). They then present a very compelling case for temporal association between a phase of increased seismicity (a preparatory factor) that culminates in two temporal clusters of large rockslides. The timing of the rockslide clusters also correlates with the occurrence of large earthquakes suggesting that earthquakes are also an important trigger mechanism. However, for their natural experiment on the role of seismicity in rockslide generation to be robust they must also show that climatic change is not a confounding variable. At some point in the discussion I expected to see Holocene climatic change excluded as a confounding variable. However, at no point is it mentioned and this is a critical issue that must be resolved before publication.

Two points make the discussion of climatic change important, these are:

1) in both lakes sedimentation rate appears to increase at about the time of the first rockslide cluster at ~4 ka. Co-temporal increases in sedimentation rate in the two lakes implies regional

forcing of which climate is a likely candidate. If climate is driving increased sedimentation rates then this might imply increased precipitation, which might play a role (along with the earthquakes) in triggering the rockslide clusters.

2) There are at least some paleoclimate records for the European Alps region that show the period between 4.5 ka and 7 ka was a time of significant climatic change (Perçoiu et al., 2017). This is the same time interval during which the authors argue that increased seismicity was the primary preparatory factor leading to rockslide initiation. Further, it appears that this is also a period of increasing storm frequency in the study region based on visual inspection of the flood deposits in Piburgersee (see supplementary figure 2). I am not familiar with European Alps paleoclimate records and only had time for a very cursory literature review, none the less some consideration of climate as a preparatory factor during this period appears necessary.

The best way to proceed would be for the authors to do a thorough literature review of Holocene climate variation in the European Alps and then provide a brief supplementary discussion outlining whether or not it could be an important contributing factor in rockslide initiation in their study area. For the authors conclusions to remain definitive they must demonstrate in this discussion that climate is not a confounding variable. A summary of this discussion should also appear in the main text. I would suggest the best place is in the Spatio-temporal relations section.

It is important to note that while the outcome of considering climatic change may or may not require the conclusions of the paper to be modified it will not impact the manuscripts novelty. The work will remain among the first attempts to evaluate preparatory and triggering factors for rockslide clusters using empirical data that spans the millennial timescales necessary for assessing these factors.

Perçoiu, A. et al. Holocene winter climate variability in Central and Eastern Europe. Scientific Reports 7, 1196, doi:10.1038/s41598-017-01397-w (2017).

Minor issues with the main text:

Line 16: given the international and general readership of the journal I do not think the term exceptionally strong earthquake is appropriate for an $ML > 5.5$. By global standards such earthquakes would be deemed in the middle of the range of moderate earthquakes. However, I recognize that these are large events by European Alps standards. Please clarify this here or remove the 'exceptionally'.

Line 17: I think that the wording here is too strong. I suggest including the words 'may have' because there is uncertainty about whether the strong earthquakes actually triggered the rockslides.

Line 51: can you please replace (>km) with (> 1 km) for clarity.

Line 98: I am really impressed by this novel analysis using CT data and the support it provides for in-situ rather than gravitational deformation. Given that SSDS 1, 5, 6, and 8 are also intraclast breccias could you please provide the rose diagrams for these in the supplement i.e. on supplementary figure 5.

Line 163: it also correlates with an increase in sedimentation rate in each lake and a period of mid Holocene climate change in the Alps that involved a switch from colder and wetter conditions associated with a more positive phase of the NAO (Perçoiu et al., 2017). Somewhere in the discussion you must deal with the potential that climate has also played a role in the triggering of the rockslides. If this is the case then it weakens your natural experiment and the ability to conclude earthquakes are a primary preparatory and triggering factor in clustered rockslide occurrence.

Line 186: I am really glad to see the relationship between rockslide location and possible epicentral location here. This independent line of reasoning is important because the chronology and hence correlations have large uncertainties. Again though, I think you need to bring the climate records into the discussion here, acknowledge that they possibly complicate the interpretation (or not) but then conclude on balance that earthquake triggering of these events is the most likely.

Line 203: should (~7ka) be a range of dates i.e. ~5.5 - 7 ka?

Line 211: towards 'a' critical state

Line 274: please briefly provide a rationale for the 0.5 cm event threshold.

Line 470: please provide the citations for the rockslides in the figure caption as you explicitly mention they are from previous work. They are best here when you first mention them rather than later on in the caption.

Minor issues with the supplement:

Line 18: could you please clarify this statement. Modified from what and how?

Supplementary figure 1: could you please change the fault line work to show the sense of motion i.e. thrust, strike slip etc.

Line 85: attesting 'to' failure

Line 101: contrasting

Lines 152-155: the sentence is worded awkwardly making the meaning unclear. I think you are saying that brittle deformation overlain by liquidized deformation caused by a single earthquake is due to the contrasting physical properties of the sediment before deformation that includes lacustrine muds overlying a debrite.

Line 180: the 'exceeding' makes the '(or more)' redundant. Personally I would say earthquakes that produce seismic intensity of VI (or more)...

Line 203: in 'the deposit'.

Line 206: amplitude 'reflectors'

Line 218-223: please reduce the complexity of this sentence to make reading it easier. The first part of the sentence appears redundant.

Oswald et al. Seismic control of large prehistoric rockslides in the Eastern Alps; response to reviewers

General

We thank the reviewers for their insightful reviews, which helped us improve our manuscript. The major concerns were the missing discussion on hydro-climatic change as possible confounding variable, the application of seismic intensity instead of other ground motion parameters and the generation mechanisms of SSDS. In the new version of the manuscript, we have revised the main text and improved the supplement to address all these concerns. In detail, we revised the main text and added the supplementary discussion 2 based on a new compilation figure on Alpine-wide hydro-climate proxies (Supplementary Figure 15), concluding that hydro-climatic change does not play a major role in triggering the rockslides of the study area. Moreover, we further improved the section on the application of seismic intensity in lacustrine paleoseismology as the most reliable method for our study area. In contrast, the use of e.g. peak ground acceleration would add further uncertainties due to the lack of knowledge on ground motion parameters of prehistorical M_L 5.5 – 6.5 earthquakes in this region. Additionally, we revised and improved the supplementary discussion 1 on SSDS, their interpretation as being formed at the lake-water surface and the possible effects of SSDS overprinting by earthquakes closely spaced in time. The latter process would make the paleoseismic record incomplete, but would not change any of the major conclusion of the paper with regard to seismic control on rockslides. We have also addressed all minor points raised by the three reviewers.

We are convinced that the manuscript got significantly sharpened thanks to the constructive comments by the reviewers and now hopefully meets the high-quality standard for publishing our best research in Nature Communications.

Our detailed comments to the reviews received on 03 September 2020 are listed below.

Many thanks and best regards,

Patrick Oswald, on behalf of all authors

We respond to the reviewer comments in blue. Line numbers refer to the annotated manuscript and supplementary information

Reviewer #1 (Remarks to the Author):

The manuscript presents support for the seismic priming and triggering of clusters of large landslides in the eastern European Alps. This support comes from the coincidence in timing of landslide clusters with evidence from two lakes of seismic disturbance of lake sediments and pulses of sediment entering the lakes. The data presented in the manuscript focuses on detailed descriptions of single sediment cores at each lake and seismic profiling provided at one of the lakes. The idea of a seismic trigger for the landslides, and therefore large prehistoric earthquakes in the Alps is not new, but this study provides strong support for this by combining two independent datasets – lacustrine and terrestrial evidence. After having identified events in both lakes that overlap in timing with dated

landslide clusters in the surrounding mountains, the authors attempt to evaluate the thresholds of seismic shaking needed to generate seismic disturbances to the lakes, and assess the likely magnitude and epicentral area of the earthquakes. They also identify earthquakes in their lake records for which there is no current evidence of landsliding, and postulate that seismic priming may be responsible for bringing slopes to a point of failure in subsequent earthquakes. The data presented appear to be robust and well explained with well-constructed figures and supporting information. The manuscript is a welcomed addition to the debate on the cause of landslide clusters in the Alps and the possibility of high-magnitude earthquakes affecting this region in the recent geological past, and will hopefully promote similar work in other localities. Below are some suggestions for how this manuscript might be strengthened:

We thank the reviewer for this positive review highlighting the importance of our work for the debate on causal factors of rockslides.

1. Given that the findings strongly suggest large magnitude earthquakes affecting the region in the recent geological past, there is perhaps a few aspects of this that could be elaborated on further by the authors. What is the societal or hazard implication of this finding – does it suggest the likely possibility of similar-sized earthquakes with the generation of large landslides in populated parts of the Alps? This is perhaps implied but it could be communicated more explicitly.

We agree that the hazard implications of the inferred earthquake-triggered rockslides in this intraplate setting do not seem to be fully developed and communicated. However, to do so in a scientifically sound manner, we first have to document recurrence patterns and understand the driving forces of the typical non-stationary seismicity in such intraplate settings, as periods of enhanced seismicity seem to be the controlling factor on catastrophic rockslides. In our records, there are too few events to conduct proper recurrence statistics for robustly assessing the probability of future M5.5-6.5 events. We addressed this comment by adding a phrase in manuscript line 247-248 to transfer the lack of our understanding on seismicity patterns in such intraplate settings.

Moreover, this paper is not the main vehicle to provide hazard implications of earthquake-prepared and -triggered rockslides, as this would be a study on its own focussing on i) quantifying processes of seismic slope destabilisation and ii) assessing and modelling current and future slope stability. For societal implications, such knowledge needs to be applied on the potential elements at risk.

Are there any existing investigations of Holocene fault surface ruptures that corroborate evidence of earthquakes in the region at the approximate dates suggested, or is the lack of surface rupturing fault traces the reason for why lake archives and landslides are needed for investigating the paleoseismicity in this region?

Exactly! The lack of any known surface rupturing fault trace in the study area is the main reason why innovative off-fault research approaches – such as the one presented here in our study – are needed for investigating paleoseismicity.

In detail, no on-fault paleoseismic studies exist in the study area and recent seismicity covers a broad region. Therefore, the source faults for the proposed paleo-earthquakes are unknown. Classic “on fault” paleoseismic research is very challenging in the region, as i) there is only low- to moderate seismicity and thus surface ruptures are typically not generated (below magnitude threshold of M5.5-6 according to Wells & Coppersmith, 1994), ii) earthquakes often occur on blind faults (Reiter et al., 2018) not producing any surface ruptures at all, iii) multiple capable faults of favourable

orientation with respect to the tectonic stress field exist. Therefore, a trench study on a single fault would only reflect a very local earthquake history, which does not represent the regional paleoseismicity. In addition, penetrative anthropogenic landscape modification of the narrow and highly-populated valleys, gravitational slope processes and high erosion rates further complicate investigations of surface ruptures. We further clarified the lack of on-fault evidence in the study area and better highlight the novelty of our study in line 59-60 of the revised manuscript .

Reiter, F., Freudenthaler, C., Hausmann, H., Ortner, H., Lenhardt, W. and Brandner, R. (2018) Active Seismotectonic Deformation in Front of the Dolomites Indenter, Eastern Alps. Tectonics, 37, 4625–4654.

Wells, D. L., & Coppersmith, K. J. (1994). New empirical relationships among magnitude, rupture length, rupture width, rupture area, and surface displacement. Bulletin of the seismological Society of America, 84(4), 974-1002.

2. The idea of seismic priming of hillslopes and the evidence for this could be tightened up or discussed further, as this is perhaps the weakest part of the manuscript.

a. It is possible that the landscape was more sensitive to earthquake-triggering in the Mid-Holocene when groundwater levels were more elevated by the climate at that time. It might be useful to discuss this point with specific reference to the papers that support a climatic link to the temporal clustering of landslides – e.g. Zerathe et al. 2014 which you cite or Panek 2019

<https://doi.org/10.1016/j.earscirev.2019.05.015> for a wider review of the topic. If such a climate influence is to be accepted then an alternative hypothesis for the reason why earthquakes did not trigger slope failures in the preceding earthquakes is that climatic conditions were more favourable for stability in those times, rather than (or in addition to) seismic priming.

We apologise to the reviewer for not discussing the potential role of climate in triggering and priming rockslides. Also under consideration of comment 2 by reviewer#3, we added a thorough discussion on this (supplementary discussion 2), based on a comparison of Alpine-wide hydro-climatic data, the rockslides of our study area and our paleoseismic data (supplementary figure 15), and summarized these in the manuscript lines 178-181 and 232-237, In detail, we now refer to Panek et al. 2019 and show that the rockslide-triggering ‘4.2 ka hydrological’ event outlined by Zerathe et al. 2014 for the SW Alps does not seem to be a significant wet period in the Eastern Alps, see supplementary discussion 2 (Line 385-390). While this additional discussion and consideration of alternative hypotheses now add scientific value and perspectives to our study, the main conclusion of the paper (i.e. earthquakes as primary process in preparing and triggering large rockslides) remains fully supported by the data and, thus, did not change compared to the initially submitted manuscript.

b. There are other studies that have investigated seismic priming that might be worth at least brief reference to. For example Parker et al. 2015 (doi:10.5194/esurf-3-501-2015) who provide support for such a process, and Brain et al., 2017 who challenge the notion that precursor seismic events always lead to weakening and priming <https://doi.org/10.1002/2016GL071879>.

We thank the reviewer for providing references supporting the process of seismic preparation. We included a reference to Parker et al. 2015 in Line 45 and in the discussion in Line 219. Brain et al., (2017) carried out geotechnical laboratory tests to simulate earthquake loading on ductile hillslope materials, where they concluded that ground shaking can in fact strengthen ductile hillslopes. We think that this is not applicable in our study area, as all slopes of the compiled rockslides are composed of either massive carbonate rocks or competent metamorphic bedrock as outlined in Line 51-52 and thus of completely different rheology as studied by Brain et al., (2017).

c. An idea for explaining the timing of paraglacial landslides long after glaciers have retreated is the stress release fracturing (static fatigue) that rock slopes may experience. For example, stress release has been postulated to have played a role in priming earthquake-triggered failures in the Scottish Highlands (Ballantyne et al., 2014; <https://doi.org/10.1016/j.quascirev.2013.12.021>). Of course, the landslide clusters in the Alps may have been a result of a combination of static fatigue and cyclic (seismic) fatigue, and both are worth considering.

Thanks to the reviewer for providing this additional reference on the static fatigue, which we implemented in Line 42.

d. Others have pointed to deep seated gravitational slope deformation (i.e gradual and potentially aseismic deformation leading to failure) influencing some of the landslide clusters in the Alps (e.g. Ostermann and Sanders 2017; <https://doi.org/10.1016/j.geomorph.2016.12.018>). It may be worth commenting on this.

Most of the deep seated gravitational slope deformations (DSGSDs) described by Ostermann and Sanders (2017) are outside of our study area, where no reliable assessments on trigger or preparation could be made based on our lacustrine datasets. Additionally, we explicitly excluded deep seated gravitational slope deformations (DSGSDs) from our compilation because DSGSDs have a complex movement history with multiple cycles of stability, creeping and acceleration phases, which are very difficult to date individually. Thus, we only focussed on short-duration mass movements originated from hillslopes with solid-rock substrates, such as rockslides and rock avalanches.

3. 3.1. The attempt to define earthquake intensity levels from the occurrence of SSDS or other sediment events is interesting and overall such an approach seems reasonable. However, I suspect that the earthquake characteristics (depth, frequency spectra, wave propagation directivity, duration) and the wave interactions with topography (e.g. amplification or de-amplification, which in turn are dependent on wave characteristics) might influence sediment response in a lake, in addition to earthquake magnitude and shaking intensity. 3.2. This could perhaps help explain the absence of sediment disturbance evidence at the time of the Köfels landslide. Perhaps some comment on such effects would be worth adding?

ad 3.1. We appreciate and fully agree with the reviewer's perspective. Earthquake ground motion characteristics are hypothesized to influence the sedimentary response and forms the focus of current and ongoing research in subduction zones settings (e.g. McHugh et al. 2020; Van Daele et al. 2019), where many recent instrumentally-recorded earthquakes of different characteristics provide quantitative constraints on the ground motion parameters. In intraplate settings of comparable lower seismicity, such as our study area, only few (if at all) significant earthquakes and respective sediment deformation occurred during the instrumental (and even historical) period. Therefore, such modern analogues are not available and evaluating the influence of ground motion on sediment deformation remains purely speculative. Thus, we use a completely different approach by calibrating the threshold at the lake site based on macroseismic intensity data of nearby villages instead of trying to quantify the exact response of the lake sediments upon seismic shaking. To date, this is the only available and commonly accepted approach for quantitative lacustrine paleoseismology (e.g. review paper; Moernaut 2020). We revised the chapter on earthquake recording threshold and added a paragraph (Supplementary Figure 6) in order to clarify this. A way forward for future research would be to conduct advanced geotechnical lab tests on undeformed natural samples, in the best way simulating the cyclic loading process on different lacustrine lithologies. However, upscaling from controlled lab tests to natural settings poses many challenges.

Ad 3.2. Hypothesizing on the Köfels case would be very speculative and cannot be supported by any of our data, as we have very little knowledge i) on the wave path-dependent ground motion parameters of such strong (prehistoric) earthquakes in the Eastern Alps, and ii) on the response of the studied lake sediments upon earthquake characteristics. We thus prefer not to address this in the present manuscript.

McHugh, C.M., Seeber, L., Rasbury, T., Strasser, M., Kioka, A., Kanamatsu, T., Ikehara, K. and Usami, K. (2020) Isotopic and sedimentary signature of megathrust ruptures along the Japan subduction margin. Mar. Geol., 428, 106283.

Van Daele, M., Araya-Cornejo, C., Pille, T., Vanneste, K., Moernaut, J., Schmidt, S., Kempf, P., Meyer, I. and Cisternas, M. (2019) Distinguishing intraplate from megathrust earthquakes using lacustrine turbidites. Geology, 47, 127–130.

Moernaut, J. (2020) Time-dependent recurrence of strong earthquake shaking near plate boundaries: A lake sediment perspective. Earth-Science Rev., 210, 103344.

Minor comments on clarity:

Main document:

5. Line 16 of abstract: consider omitting the word ‘exceptionally’. $M_L > 5.5$ is probably not exceptionally strong...though I do live on an active plate boundary so perhaps I have a different tolerance ;-). Consider also omitting the word ‘ultimately’ on the same line.

We are fully aware that earthquakes with $M_L > 5.5$ are only moderate in a global view. However, due to the shallow focal depth (5-10km), a $M_L > 5.5$ would reach high intensities ($\geq VIII$). We addressed this comment by changing the adjective to “severe” earthquakes in the abstract in Line 16 (and later in the manuscript). “Severe” has a direct connotation to the seismic ground shaking (i.e. seismic intensity), which is much more important in our investigation than the total energy release of the earthquakes (i.e. magnitude). We also kindly refer to the response of comment 9 by reviewer 2.

We added some words on L202 to highlight the comparably higher magnitude ($M_L 5.5-6.5$) of the newly reported paleo-earthquakes in comparison to the documented historical earthquakes in the region: In the broader study area so far only five events $M_L 5-5.3$ were documented in the last ~400 years. Moreover, we improved the explanation that this moderate magnitude earthquakes generate severe shaking in the Eastern Alps and are heavily damaging in the introduction in Lines 58-60. We changed the term ‘strong earthquakes’ in the whole main text and supplement to either ‘severe earthquakes’ or ‘severe seismic shaking’.

6. Line 17 of abstract: perhaps add the words ‘are likely to have’ to this sentence to help convey the uncertain nature of this statement.

We added ‘is likely to have’ in Line 18.

7. Line 20 of abstract: it is unclear what is ‘new’ about the methods presented. Again on L67-68 you imply that this research is novel in using lake sediments to evaluate the causes (i.e. earthquake triggers) of landslides. This is not really true. Perhaps what is novel is the integration of these two datasets in the eastern Alps.

We addressed this comment by changing the 'new method' to 'novel application of the new lacustrine paleoseismic records' in Line 21-22 and by also revising the sentence in Line 72-73.

Supplementary Information document:

8. Line 145-147. This sentence is unclear. Are you saying that the geochemistry of the SSDS is similar to the underlying undeformed sediments? Also if this is the intent, it is hard to see why this is considered evidence for the deformation to have occurred at the sediment-water interface. If the deformation occurred below the sediment-water interface it presumably would also share the same geochemistry as any adjacent undeformed sediments.

We thank the reviewer for pointing out this confusing sentence. We thoroughly revised the whole Supplementary Discussion 1 to straighten the argumentation line even more. In this specific case, we took apart the argumentations on i) SSDS generating at the water-sediment interface (supplement Line 147-153) from ii) the argumentations that SSDS are in-situ (supplement Lines 153-163) and not the product of gravitational flow processes.

9. L152-155. This sentence is a bit hard to follow. Consider revising.

Thanks for hinting at this confusing sentence, which we changed in Lines 176-179 in the course of a thorough revision of Supplementary discussion 1.

10. L215. Should be 'extent' not 'extend'

We addressed this comment in Line 263.

11. Supplementary Figure 9: What are the red triangles on the map of Event Horizon G?

The red triangles are a left-over of a previous version of the figure, which we oversaw to remove in event horizon G. The figure is now updated and all triangles are removed.

Yours sincerely,
S McColl

Reviewer #2 (Remarks to the Author):

I really enjoyed reading the manuscript. This is a nice piece of multidisciplinary research in geosciences bringing together, lake sedimentology, landslides and seismicity. The figures are clear, the text is very well readable. However, there are several remarks and amendments marked in the annotated manuscript attached.

We thank the reviewer for these positive statements highlighting the multi-disciplinary nature of our research.

1.1. Starting with the most critical issue: the earthquakes and magnitudes.

The 1930/10/08 eq was deep, up to 35 km are estimated and had a thrust mechanism. Those deep

eqs do generally not produce surface ruptures, as they are "blind" to the surface. You do not mention the date, and that there was heavy rainfall (after, but unknown before).

1.2. So, if there are no surface evidence, and little deformation described (some rockfalls, but no major severe landscape changes), how can you find the "postseismic landscape changes" in your archives? I have the feeling this is over-interpretation, also you do not provide alternative thoughts?

Ad 1.1: The focal depth of 36 km \pm 9 km after estimations from Gräfe (1933) has been revised by Franke and Gutdeutsch (1973) to 8 km depth. Nowadays, Austrian seismologists from ZAMG (Central Institute for Meteorology and Geophysics) provide a best fitting estimate to 9 km using macroseismic intensities (AEC, 2020). This hypocentral depth (9 km) is in accordance with the relatively shallow current seismicity (5-10 km) according to Reiter et al. 2018, which was outlined in the manuscript Line 56-58 and Supplementary Figure 1.

The exact date (1930-10-07 / 11:27 p.m.) is provided in supplementary table 7. The heavy rainfall the day after the earthquake might have contributed in triggering one or the other of the small-scale gravitational mass movements documented by Krauss (1932) shown in Supplementary Figure 6, but were not further used in our study for any argumentation on postseismic landscape response.

Ad 1.2: This must be a misunderstanding by the reviewer. We do not document any postseismic landscape response in Plansee for the CE 1930 Namlos earthquake. Postseismic landscape response is only reported from our sedimentary archives for the largest inferred earthquakes of the region (events C and E in Plansee record, see also Supplementary Figure 12).

The catchment of Plansee is built up by steep cliffs consisting of a densely jointed dolomite succession (Hauptdolomit) covered by numerous talus slopes and debris-filled minor valleys with non-permanent rivers (mentioned in Supplementary Note 1). Each extreme-discharge event generates a clastic bed intercalated in the background sediment (see Supplementary Figure 12). The the post-event period of event C and event E exhibit striking differences to this pattern: i) the number of clastic deposits is abnormally high in the immediate aftermath of the event and decreases in the next years/decades and ii) background sediment between the clastic layers is quasi-absent and only transits back to normal within decades, as described in Supplement Lines 307-314 of Supplementary Figure 12.

R. Gräfe: "Das Nordtiroler Beben vom 8. Oktober 1930." Zeitschr. f. Geoph., Bd. B u. 9 (Braunschweig 1932 u. 1933).

Franke, A. & Gutdeutsch, R. 1973. Eine makroseismische Auswertung des Nordtiroler Bebens von Namlos am 8. Oktober 1930. Mitteilungen der Erdbeben-Kommission, Neue Folge Nr. 73, Österr. Akad. d. Wissenschaften, Vienna, Austria.

Kraus, E. (1932) Die Bewegung des Erdbebens am 8. Oktober 1930 im süddeutschen Bau.

2. Also, please explain a 2 m thick "turbidite" in the archive, though it is really a small archive? Where does the sediment come from? Explain better, as the other layers are max. 35 cm thick. Even megaturbidites in deep water hardly form that thick layers.

We apologise to the reviewer for not better explaining the extraordinary thick turbidite. We now explain that the 2.5 m thick turbidite was induced by a local rock fall impacting and remobilizing the sediments in the lake basin (Supplementary Figure 4c, Lines 109-111). The large volume is generated through remobilization and resuspension of the lake floor sediments upon impact of the rock mass,

which is evidenced by the dominant lithology of the turbidite (organic-rich mud) and the presence of a morphological depression and compressional ridge related to the impact process.

3. secondary is not cascading!

Thank you for hinting at this flaw, which we corrected by adding 'and' in Line 27.

4.- climatic deterioration - evidence and what does this mean ? It is rather subjective, I would say: more rain, colder, more snow, less precipitation.....

We changed "climatic deterioration" to "hydro-climatic change" in the whole manuscript. Moreover, we added evidence and explanation in supplementary discussion 2 and added a sentence in the manuscript Line 178-181 and Lines 232-237 following the constructive comments of reviewer #3 and #1.

5. 5.1. The distance or proximity to active faults (and not only epi/hypocentres) is important, I did not find that? Are these faults capable for the eqs needed? 5.2. Why put it in the center of the "transect"? Makes no sense to me, and is just a game to play.

Ad 5.1: The active faults that have ruptured during the paleo-earthquakes are unknown. We agree that the distance between the lakes and the earthquake is important, however, given the fact that the earthquake-causing fault remains unknown, one only can discuss plausible or minimum-requirement scenarios based on off-fault paleoseismic evidence (see method of Kremer et al. 2017). Potentially active faults have been identified by Reiter et al., 2018 and some of these faults are blind (also shown in figure below from Reiter et al. 2018). The faults are capable of producing M_L 6.5 earthquakes according to focal-depth distributions inferred from macroseismic data (Lenhardt et al., 2007) as outlined in manuscript in Line 199-200.

Ad 5.2: When only a few sites with off-fault paleoseismic data is available, the common approach is to calculate earthquake scenarios in terms of magnitude and location based on intensity predication equations (see Kremer et al. 2017, Strasser et al., 2006). Our approach is to estimate the minimum magnitude for an earthquake that can generate the observed sedimentary evidence in both lakes, as outlined in Line 195-197 and explained in Supplementary Method 1. Logically, given our simple model, this hypothetic smallest earthquake would be located in the middle between both lakes. Every change in epicentral location for such a hypothetical minimum-scenario would lead to a higher magnitude and is no minimum estimation anymore. For the upper bound of the magnitude range, we use the value M_L 6.5 derived from focal-depth distributions inferred from macroseismic data (Lenhardt et al. 2007). Moreover, when projecting our hypothesized epicentre into the cross section of the figure from Reiter et al. 2018 below, the epicentre is located within the high seismicity area and is therefore considered a plausible scenario with respect to the current-day seismotectonic activity (see also black star in cross section below representing the projected location of such a hypothetical scenario). We added a sentence in the supplementary method 1 (Lines 454-455) to clarify that the hypothetical epicentre location is reasonable as it actually lies within an area of high recent seismicity according to Reiter et al., 2018.

Kremer, K., Wirth, S.B., Reusch, A., Fäh, D., Bellwald, B., Anselmetti, F.S., Girardclos, S. and Strasser, M. (2017) Lake-sediment based paleoseismology: Limitations and perspectives from the Swiss Alps. Quat. Sci. Rev., 168, 1–18.

Strasser, M., Anselmetti, F.S., Fäh, D., Giardini, D. and Schnellmann, M. (2006) Magnitudes and source areas of large prehistoric northern Alpine earthquakes revealed by slope failures in lakes. *Geology*, 34, 1005.

Reiter, F., Freudenthaler, C., Hausmann, H., Ortner, H., Lenhardt, W. and Brandner, R. (2018) Active Seismotectonic Deformation in Front of the Dolomites Indenter, Eastern Alps. *Tectonics*, 37, 4625–4654.

Lenhardt, W.A., Freudenthaler, C., Lippitsch, R. and Fiegweil, E. (2007) Focal-depth distributions in the Austrian Eastern Alps based on macroseismic data. *Austrian J. Earth Sci.*, 100, 66–79.

Figure 1: Cross section with focal mechanisms of recent earthquakes of the study area by Reiter et al. (2018). Note that some fault activity is blind, shallow aseismic or slowly moving (blue). Black star represents the potential location for the minimum magnitude calculation outlined in Supplementary Method 1.

6. The area is really not the "most seismic area in the E. Alps), by far not. No M 7 is ever found in the Alps (maybe in Vienna Basin and Friuli).

This quote is not correct. We wrote "one of the most seismically active areas in the Eastern Alps" (Line 54-56), which is already clear by looking at a seismic hazard map:

<https://www.gfz-potsdam.de/en/section/seismic-hazard-and-risk-dynamics/projects/previous-projects/probabilistic-seismic-hazard-assessments/d-a-ch-seismic-hazard-for-the-d-a-ch-countries-germany-austria-switzerland/>

7.1. You never discussed, because you relate 1 SSDS to 1 eq, but is this true?

7.2. Can a large eq overprint older structures? 7.3. Can an eq cause SSDS in several of your layers? Why not? 7.4. I can see huge differences in the deformation style, e.g semi-brittle deformation and even microfaults? So, this layer was "harder" more rigid to deformation? Discuss this please. And this also effects your age discussion and clustering. Discuss this as well.

We apologise to the reviewer for not explaining the generation and nature of SSDS in sufficient detail and we acknowledge these complementary ideas, which we included and further discussed in the supplementary discussion 1.

Ad 7.1: For the event layer SSDS 2, two deformation structures are described and interpreted to be co-genetic. We revised and clarified the explanation also based on the comment of reviewer#3 in the supplementary discussion 1 in Lines 176-179. For the other SSDS we changed some wordings in Supplementary Discussion 1 (Lines 149-151) to be more precise.

Ad 7.2: We appreciate this comment by the reviewer and added a small paragraph in the supplementary discussion 1 in Line 164-170, where we now explain the potential incompleteness of a SSDS-derived paleoseismic archive due to potential overprinting of a subsequent earthquake. This would not change the interpretation of an SSDS to be earthquake-related and would only add even more earthquakes to the paleoseismic archive, especially within the period of enhanced seismicity.

Ad 7.3: Co-genetic development at several stratigraphic levels can be excluded based on sedimentological observations indicating e.g. gravitational settlement process (normal grading) or directly over- and underlying undeformed sediment. We revised and strengthened the whole paragraph in the supplementary discussion 1 to clarify the generation of SSDS at the water-sediment interface. Specifically, the co-genetic development is further explained in Line 147-153.

Ad 7.4: The observation of different deformation styles showing a sequence in deformation style makes it very comparable to what is documented in the well-studied Lisan Formation in the Dead Sea area and which was numerically modelled following the process of Kelvin-Helmholtz Instability (Wetzler et al. 2010). The occurrence of microfaults is now further explained in lines 171-179 of the supplementary discussion 1 and also mentioned in the comment to point 7.1.

Wetzler, N., Marco, S. and Heifetz, E. (2010) Quantitative analysis of seismogenic shear-induced turbulence in lake sediments. Geology, 38, 303–306.

8. The next problem are sedimentation rates, between the 1930 and the present deposits are c. 25 cm in 90 years, but the date before is c. 500 BP and sediment accumulated also 25 cm? Explain variation in rates.

In lacustrine sediment cores it is common that the uppermost sediment shows much higher sedimentation rates mainly due to high water content close to the sediment-water surface and increase in compaction with sediment depth (see also age-depth model of Piburgersee in Supplementary Figure 3). In addition to that, Plansee is dammed and has been used as a hydropower reservoir since the begin of 20th century, leading to seasonal lake level fluctuations of about five meters since then. This led to enhanced erosion of coastal sediments and thus also increased sedimentation rate. We added explanation on this in Supplementary Note 1 (line 35-36), Supplementary Figure 3 (lines 72-73) and Supplementary Figure 11 (line 294-298) to clarify and explain the increased sedimentation rate at the top of the core.

9. You are citing much larger eqs for the "postseismic sediment flux" and landscape response, like the Alpine fault (M 7.6, which means c. 1000 times more energy was released; the Turkish ones are min.

5.5, but slightly higher up to 6.5) - is this a wishful thinking? Re-check and provide (much) better arguments.

The proposed magnitude range in this study (M_L 5.5-6.5) lies within the magnitude range for which earthquake response is documented in SW-Turkey Avşar et al. 2016).

Nevertheless, the magnitude is not a relevant parameter whether catchment response occurs or not; what matters is the strength of the seismic shaking. A shallow M_L 5.5 earthquake in an intraplate setting can have a similar epicentral intensity as a deeper higher magnitude earthquake. We have made this distinction between magnitude (energy release) and intensity more clear in the revised manuscript by explaining why these earthquakes with globally only moderate magnitudes are actually severe in this region in Line 57-60.

Avşar, U., Jónsson, S., Avşar, Ö. and Schmidt, S. (2016) Earthquake-induced soft-sediment deformations and seismically amplified erosion rates recorded in varved sediments of Köyceğiz Lake (SW Turkey). J. Geophys. Res. Solid Earth, 121, 4767–4779.

10. More in the annotated ms.

Minor comments in the annotated manuscript were addressed and changed using track change, where applicable in accordance with the general comments above. Following comments were digitised from the annotated MS and explained separately:

11. Line 16, 17, 20: “evidence”

We are fully aware, that in the abstract some “evidence” might be missing, but we have to follow the editorial guidelines with a limited word count of ~150 for the abstract. However, for the three mentioned points in the text of missing evidence, we gave a complete argumentation line up in the main manuscript at Lines 197-202, 211-222 and 164-167

12. Line 106: “Why not loading → normal faults”

Loading can be excluded, as there is no evidence for “rapid” sedimentation above this SSDS which could generate a sudden increase in overburden stress. We further explained this in Supplementary Discussion 1 Line 171-176.

13. Line 109-116: “This is more discussion/conclusion → I miss an age discussion.”

This is our method/research approach that leads to the evaluation of triggering of rockslides. We don't quite understand what the reviewer means with age discussion. If it were to be about earthquake overprint of SSDS: we address this now in Supplementary Discussion 1 (Lines 164-170), as replied to comment 7; or if it were to be about the numbers in Figure 2: they reflect ^{14}C ages, which is explained in the figure caption. In any case, our research approach is not something to be moved to the discussion part of the main manuscript.

14. Line 159: “? moderate and not larger than historical”

We postulated that some of the earthquakes are stronger than the historical CE 1930 earthquake based on sedimentological observations outlined in Lines 114-118 for Piburgersee and in Lines 155-160 for Plansee. For further clarification we explicitly mention now in the manuscript in Line 202-203 that “(M_L 5.5-6.5) are larger than historically-documented events in the study area (M_L up to 5.3)”.

15. Line 194: “I think more important are peak ground amplifications ~ 0.5 g???”

We appreciate this comment of the reviewer, as peak ground acceleration (PGA) indeed may be an important parameter for the generation of SSDS (see e.g. Avşar et al. 2016) and is a more

quantitative parameter than intensity. However, the input data for our analysis are the documented intensity data points of historical earthquakes, which are used to calculate seismic intensity at the lakes, to define shaking thresholds for sedimentary evidence and to estimate the minimum magnitude of the paleo-earthquakes based on intensity prediction equations. We prefer to hold to the primary data (i.e. intensity) against which our lake sites have been thoroughly calibrated (see detailed and revised version of supplementary figure 6) instead of introducing significant extra uncertainty by going from IDPs to earthquake magnitude and focal depth and then via a ground motion prediction equation (derived from smaller earthquakes) to a PGA value. We revised the chapter on earthquake recording threshold and added a paragraph (Supplementary Figure 6) in order to clarify the use of intensity data.

We here also kindly refer to the detailed explanation (and respective revision in the manuscript and supplementary data to substantiate this) outlined in response to reviewer 1 comment 3.1 (above) justifying the approach of calibrating against macroseismic intensities, and why – for intraplate settings – this approach for quantitatively constraining earthquake shaking threshold values beyond historically calibrated macro-seismic intensity values is state-of-the-art. However, future research should aim at more quantitatively constraining the ground motion parameters of such large prehistoric earthquakes.

Avşar, U., Jónsson, S., Avşar, Ö. and Schmidt, S. (2016) Earthquake-induced soft-sediment deformations and seismically amplified erosion rates recorded in varved sediments of Köyceğiz Lake (SW Turkey). J. Geophys. Res. Solid Earth, 121, 4767–4779.

16. Line 445: “these are thrust (planes) not faults! → inactive tectonic windows.”

Although we are confused by this comment on terminology of thrust (planes) and faults, we can imagine that the reviewer is hinting that not all of the thrust faults belonging to the “Lechtal thrust” can be recently active, as the Cretaceous thrust plane has been folded in the Paleogene and is outcropping as tectonic windows. This is true except for the southernmost fault, as this is connected to the base of the Northern Calcareous Alps (see also blue to black fault outcropping slightly below dwelling “Vorderriss 1” in cross section of Reiter et al. 2018 provided in reply to comment 5). Besides its recent inactivity (shown by Reiter et al. 2018 by the blue coloring) it can be potentially active in the current N-S stress field. This is irrelevant anyway, as we never in this manuscript link the paleo-earthquakes to any potentially capable faults and only provide the faults for a complete geological overview. However, we have clarified the existence of tectonic windows by adding more kinematic indicators to the faults and revising manuscript figure 1.

17. Line 453: “Does starting lithification produces fractures? Or breccia?”

If this would be the case, we would observe it in each flood layer, which is not the case. To our knowledge, a normal consolidation process in a sedimentary sequence in a flat basin does not alter the lamination integrity in the considered subsurface depths. This is evidenced by many paleoclimate studies on Holocene varved (i.e. annual laminated) records that show intact laminations (e.g. Czymzik et al. 2013).

Czymzik, M., Brauer, A., Dulski, P., Plessen, B., Naumann, R., von Grafenstein, U. and Scheffler, R. (2013) Orbital and solar forcing of shifts in Mid- to Late Holocene flood intensity from varved sediments of pre-alpine Lake Ammersee (southern Germany). Quat. Sci. Rev., 61, 96–110.

The methods (geophysical and sedimentological, dating etc.) are excellent.

We thank the reviewer for this very positive comment.

Abstract:

17.1 I can not see any clustering in the eq data? 17.2 You claim for $M > 5.5$ but there is no evidence, neither historical nor instrumental provided. 17.3 You speak of series of "strong" eqs. For my feeling strong is larger than $M 6.5$. Your eqs are moderate, think of the energy released. And also think of the focal depth. 17.4 You never discussed peak acceleration values? Why those are important for triggering landslides, but again it is a function of depth. I guess your eqs are around 0.4-0.5 g, so far less in comparison to what has been observed in Norcia or Finale Emilia (blind as well).

17.5 Also, I have problems of the "weakened" rock masses prone to fail...(because of repeated eqs). What about changing weathering rates or post-LGM (deglaciation debris?).

ad 17.1: In Line 164-166 and Figure 4 we show the enhanced seismicity period ('earthquake cluster') with average recurrence rates of few 100 years between 7 to 3 ka BP in the southern study area, whereas there is seismically quiescence for several millennia before (9.9 to 7 ka BP) and after (3 ka BP to CE1930).

Ad 17.2: We revised the potential magnitude range of prehistoric earthquakes to $M_L 5.5-6.5$ in Line 16-17. We further explained the method and use of this also in replies to comment 5.2. and 11. The $M_L 5.5 - 6.5$ is derived from a minimum magnitude estimation based on our hypothetical earthquake calculation in the middle between both lakes (see supplementary method 1) and the upper magnitude bound is derived from focal-depth distributions inferred from macroseismic data (Lenhardt et al., 2007).

Ad 17.3. We thank the reviewer for pointing out the globally wrong nomenclature used for these 'moderate' earthquakes. We addressed this comment by changing the terms 'strong earthquakes' to 'severe earthquakes' or 'severe seismic shaking' further outlined in reply to reviewer#1 comment 5. It is not about the energy released at the rupture area, but about seismic intensity at the considered site, which (besides mainly magnitude and travel distance) is also a function of the focal depth as mentioned by the reviewer himself.

Ad 17.4. We addressed this comment in reply to comment 14 and added a paragraph in the supplementary figure 6 on this. PGA probably is an important parameter, but its use would introduce much more uncertainty in our analysis due to parameter conversions.

Ad 17.5. Seismic fatigue of rock slopes is a process which has been speculated (Prager et al. 2008) and so far only numerically modelled (Gischig et al. 2016), as direct investigations "are difficult due to long reoccurrence time and the unpredictability of strong severe earthquakes" as outlined in the manuscript line 45-47. Nevertheless, Gischig et al. 2016 showed that earthquake-induced generation and propagation of new fractures weakens a relatively strong rock mass. We also introduce the role of hydromechanical fatigue (Preisig et al. 2016) as rock mass weakening process in Line 44. In addition, we added a whole discussion on the effect of Holocene hydro-climatic change on the rockslides in the study area (supplementary discussion 2) and conclude now that it is only a subordinate mechanism in generating these rockslides in the manuscript Line 232-237.

Gischig, V., Preisig, G. & Eberhardt, E. Numerical investigation of seismically induced rock mass fatigue as a mechanism contributing to the progressive failure of deep-seated landslides. Rock Mech. Rock Eng. 49, 2457–2478 (2016).

Prager, C., Zangerl, C., Patzelt, G. & Brandner, R. Age distribution of fossil landslides in the Tyrol (Austria) and its surrounding areas. Nat. Hazards Earth Syst. Sci. 8, 377–407 (2008).

Preisig, G., Eberhardt, E., Smithyman, M., Preh, A. and Bonzanigo, L. (2016) Hydromechanical rock mass fatigue in deep-seated landslides accompanying seasonal variations in pore pressures. Rock Mech. Rock Eng., 49, 2333–2351.

Videos: add scale.

We apologise for not adding a vertical scale to the videos, which we now changed in the revised videos.

The manuscript needs major revisions, as too many evidences and arguments are around the threshold value during eq shaking. And some interpretations are not sound.

But as the ms is interesting and novel, the ms should be elaborated better.

regards

We are convinced that we now address all remarks by reviewer 2 in the new version of this manuscript and supplement. For the few cases where we were confused by the comments of the reviewer, we took these as an opportunity to rewrite and further clarify the text to reduce future misunderstandings.

Klaus Reicherter, RWTH Aachen University

Reviewer #3 (Remarks to the Author):

To the editor and authors,

The manuscript title ‘Seismic control of large prehistoric rockslides in the Eastern Alps’ submitted to Nature Communications by Oswald et al. uses lacustrine paleoseismology to explore the link between seismicity and the occurrence of large rockslides in the European Alps. The authors present lacustrine paleoseismic reconstructions of unusually high quality. More importantly, they apply them in a novel way to test long standing hypotheses about seismicity as both a preparatory and a triggering factor in the initiation of large rockslide clusters. The authors use their impressive lacustrine paleoseismology dataset to argue that seismic fatigue is the main preparatory factor causing clustering of large rockslide occurrence in the European Alps. The work is suitable for publication in Nature Communications for three main reasons: i) the work presents very high quality lacustrine paleoseismology datasets; ii) these high quality datasets are applied in a very novel way to address a scientific question that has remained poorly tested until now; and iii) both the results and approach have global applicability because they elucidate the processes that control large rockslide initiation and hence have relevance for landscape evolution and hazard in mountain landscapes.

The manuscript is very well written and the authors have done an excellent job of compiling a detailed supplement for the specialist reader. It is my opinion that this is the perfect set up for a Nature Communications paper. Generally, all the critical information for a general reader to understand the arguments developed in manuscript is present in the main text, while the supplement provides the detail required by the specialist reader who wants to mine deeper in to the dataset that supports the author’s conclusion.

While I commend the authors for their attention to detail and ingenuity, I have one major concern that must be addressed before the manuscript is accepted for publication. Therefore, I recommend

minor/moderate revisions before the manuscript is accepted for publication. I have also identified a series of minor issues that are outlined below, referenced by line number.

Thank you for this very positive review acknowledging the high data quality and its novel application, but also for providing extensive comments and suggestions on the major concern 'climate change as confounding variable'. We are convinced that we now fully address this concern in a dedicated supplement and several lines in the manuscript as well as all the other minor issues by reviewer 3.

Kind regards,

Anon.

Major concerns:

Climatic change as a confounding variable in the author's natural experiment.

In their introduction the authors acknowledge that both seismicity and climatic change are often invoked to explain temporal clustering of large rockslides (line 33). They then present a very compelling case for temporal association between a phase of increased seismicity (a preparatory factor) that culminates in two temporal clusters of large rockslides. The timing of the rockslide clusters also correlates with the occurrence of large earthquakes suggesting that earthquakes are also an important trigger mechanism. However, for their natural experiment on the role of seismicity in rockslide generation to be robust they must also show that climatic change is not a confounding variable. At some point in the discussion I expected to see Holocene climatic change excluded as a confounding variable. However, at no point is it mentioned and this is a critical issue that must be resolved before publication.

We apologise to the reviewer for not discussing the role of climate in potentially priming and triggering rockslides. This is now addressed in a dedicated supplementary discussion 2, supplementary figure 15 and several sentences in the main manuscript (lines 178-181 and Lines 232-237) as also outlined in the response to reviewer 1 comment 2. After compiling available hydro-climatic proxy data, we compared the hydro-climate reconstructions with rockslide ages and our paleoseismic record and complementing with scientific discussions with the paleoclimate expert of our institute, Prof. Christoph Spötl. As a result of this comparison, we can conclude that extreme precipitation events or enhanced total precipitation periods can be reasonably ruled out as trigger mechanisms for generating the two rockslide "events" (3.0 ka BP and 4.1 ka BP) in the study area. Thus, the main conclusion of the paper (i.e. earthquakes as primary process in triggering large rockslides) remains fully supported by the data and did not change compared to the initially submitted manuscript.

Two points make the discussion of climatic change important, these are:

- 1) in both lakes sedimentation rate appears to increase at about the time of the first rockslide cluster at ~4 ka. Co-temporal increases in sedimentation rate in the two lakes implies regional forcing of which climate is a likely candidate. If climate is driving increased sedimentation rates then this might imply increased precipitation, which might play a role (along with the earthquakes) in triggering the rockslide clusters.

The reviewer is correct by observing a change in sedimentation rate in both lakes around 3.8 ka BP, pointing to a regional forcing. However, sedimentation rates actually were significantly lower (not higher) between ~3.8 ka and ~2.0 ka BP, with a decrease from 0.045 to 0.020 cm/a in Plansee and from 0.025 to 0.014 cm/a in Piburgersee (Supplementary figures 3 and 11; Supplement lines 397-400). This may relate to a decrease in clastic input, typically related to a drier climate or to a reduced availability of sediment in the catchment (less weathering?). Moreover, the flood record of Piburgersee, which we now also attached to the supplementary figure 15 shows fewer flood events during the low sedimentation rate period 3.8-2.0 ka BP. Although the driving force behind this decrease in sedimentation rates is unknown, it contradicts the idea of increased precipitation in this period and therefore, we implemented this observation on sedimentation rates in the supplementary discussion 2 on hydro-climate change as triggering mechanism of rockslides (Supplementary Discussion 2, Lines 397-400).

2) There are at least some paleoclimate records for the European Alps region that show the period between 4.5 ka and 7 ka was a time of significant climatic change (Peroiu et al., 2017). This is the same time interval during which the authors argue that increased seismicity was the primary preparatory factor leading to rockslide initiation. Further, it appears that this is also a period of increasing storm frequency in the study region based on visual inspection of the flood deposits in Piburgersee (see supplementary figure 2). I am not familiar with European Alps paleoclimate records and only had time for a very cursory literature review, none the less some consideration of climate as a preparatory factor during this period appears necessary.

The best way to proceed would be for the authors to do a thorough literature review of Holocene climate variation in the European Alps and then provide a brief supplementary discussion outlining whether or not it could be an important contributing factor in rockslide initiation in their study area. For the authors conclusions to remain definitive they must demonstrate in this discussion that climate is not a confounding variable. A summary of this discussion should also appear in the main text. I would suggest the best place is in the Spatio-temporal relations section.

We highly appreciate this suggestion and followed it by a thorough literature review on hydro-climate data resulting in supplementary figure 15 and by adding a dedicated supplementary discussion 2 and several sentences in the manuscript (lines 178-181 and Lines 232-237). As mentioned by the reviewer, we also added the Piburgersee flood record to the Supplementary figure 15. In conclusion, Holocene hydro-climate variability in the Holocene can be expected to likely play a role in long-term failure preparation due to subcritical fracture growth and hydromechanical fatigue. However, enhanced precipitation on its own can be reasonably ruled out as triggering mechanism of these rockslides, because i) there is no climate influence documented for the rockslides Fernpass and Eibsee at 4.1 ka BP, although there is a hydrological event at 4.2 ka BP documented in the Southern and Western Alps, which is interpreted to be the trigger of rockslides in the SW Alps (Zerathe et al, 2017) , ii) no large rockslides occurred in the study area during the wettest periods in the Holocene e.g. the Little Ice Age, and iii) the majority of the rockslides in the study area occurred in the period 4.4-3.0 ka BP which does not coincide with an inferred Alpine-wide nor with a regional wet period in the Eastern Alps. In contrast, a striking spatiotemporal relationship exists between paleo-earthquakes and rockslides Fernpass and Eibsee at 4.1 ka BP and Tschirgant and Haiming at 3.0 ka BP inferring that severe earthquakes were the main trigger mechanism of these rockslides.

Zerathe, S., Lebourg, T., Braucher, R. and Bourlès, D. (2014) Mid-Holocene cluster of large-scale landslides revealed in the Southwestern Alps by ^{36}Cl dating. *Insight on an Alpine-scale landslide activity. Quat. Sci. Rev.*, 90, 106–127.

It is important to note that while the outcome of considering climatic change may or may not require the conclusions of the paper to be modified it will not impact the manuscripts novelty. The work will remain among the first attempts to evaluate preparatory and triggering factors for rockslide clusters using empirical data that spans the millennial timescales necessary for assessing these factors.

Perşoiu, A. et al. Holocene winter climate variability in Central and Eastern Europe. *Scientific Reports* 7, 1196, doi:10.1038/s41598-017-01397-w (2017).

Thank you for this very positive statement! The conclusion of the paper does not need to be modified, because we could show that hydro-climatic change is not a confounding variable for earthquake triggered rockslides of the Eastern Alps. (see reply to previous comments and supplementary discussion 2 and summary sentences in manuscript lines 178-181 and 232-237). We also thank the reviewer for providing the additional reference, which we decided not to use: Perşoiu et al 2017 investigates winter precipitation in an ice cave in the Romanian Carpathian Mountains, which is far east of our study area. However, we added some precisely dated regional and Alpine hydroclimatic datasets to our comparison of climate with rockslides in Supplementary Figure 15 and discussed them in Supplementary Discussion 2.

Minor issues with the main text:

Line 16: given the international and general readership of the journal I do not think the term exceptionally strong earthquake is appropriate for an $M_L > 5.5$. By global standards such earthquakes would be deemed in the middle of the range of moderate earthquakes. However, I recognize that these are large events by European Alps standards. Please clarify this here or remove the 'exceptionally'.

We apologise this misuse of global terminology. We changed the wording on 'strong earthquakes' to 'severe earthquakes' or 'severeseismic shaking' in the whole manuscript, as this is the correct terminology for a seismic intensity of VIII. Further, the " $M_L > 5.5$ " is now changed to the magnitude range 5.5-6.5 based on our minimum magnitude estimations and the maximum possible magnitude based on focal-depth distributions inferred from macroseismic data (Lenhardt et al., 2007). Please see also the detailed replies to reviewer 1 comment 5 and reviewer 2 comment 17.3.

Lenhardt, W.A., Freudenthaler, C., Lippitsch, R. and Fiegweil, E. (2007) Focal-depth distributions in the Austrian Eastern Alps based on macroseismic data. *Austrian J. Earth Sci.*, 100, 66–79.

Line 17: I think that the wording here is too strong. I suggest including the words 'may have' because there is uncertainty about whether the strong earthquakes actually triggered the rockslides.

We addressed and changed to 'is likely to have' in Line 18 following the suggestion of reviewer #1 in comment 6.

Line 51: can you please replace (>km) with (> 1 km) for clarity.

We addressed this by changing the rockslide run-out distance to the correct value 'up to 16 km' in Line 53

Line 98: I am really impressed by this novel analysis using CT data and the support it provides for in-

situ rather than gravitational deformation. Given that SSDS 1, 5, 6, and 8 are also intraclast breccias could you please provide the rose diagrams for these in the supplement i.e. on supplementary figure 5.

We thank the reviewer for this positive statement on our method using CT data. We included now all rose diagrams in the Supplementary Figure 5 and referred to them in the manuscript Line 103

Line 163: it also correlates with an increase in sedimentation rate in each lake and a period of mid Holocene climate change in the Alps that involved a switch from colder and wetter conditions associated with a more positive phase of the NAO (Perşoiu et al., 2017). Somewhere in the discussion you must deal with the potential that climate has also played a role in the triggering of the rockslides. If this is the case then it weakens your natural experiment and the ability to conclude earthquakes are a primary preparatory and triggering factor in clustered rockslide occurrence.

Sedimentation rates are actually decreasing (please also see reply to comment 1; Supplementary figures 3 and 11). In the discussion of the main text we added several sentences concluding that climate played a subordinate role in triggering the investigated rockslides in Lines 178-181 and 232-237 based on a new supplement discussion and a comparison of hydroclimate with rockslides and our paleoseismic record (supplementary figure 15). We kindly refer to response of comment 2 for a conclusion on the detailed reasoning why climate can be reasonably excluded as triggering mechanism.

Line 186: I am really glad to see the relationship between rockslide location and possible epicentral location here. This independent line of reasoning is important because the chronology and hence correlations have large uncertainties. Again though, I think you need to bring the climate records into the discussion here, acknowledge that they possibly complicate the interpretation (or not) but then conclude on balance that earthquake triggering of these events is the most likely.

We addressed this comment by adding the sentence in lines 178-181 and line 232-237. We kindly refer to the previous comments for a conclusion on the subordinate role of hydroclimate in triggering the rockslides in the study area and to supplementary discussion 2 and supplementary figure 15 for a detailed discussion on this topic.

Line 203: should (~7ka) be a range of dates i.e. ~5.5 - 7 ka?

The period of enhanced seismicity initiates at 7 ka BP and lasts until 3 ka BP (Figure 4). We removed parentheses and added 'at 7 ka BP' in Line 212 to further clarify the timing of initiation.

Line 211: towards 'a' critical state

Thanks for noting this issue, which we fixed now in Line 222.

Line 274: please briefly provide a rationale for the 0.5 cm event threshold.

0.5 cm represents the threshold where one can macroscopically identify event layers with high confidence. This threshold is more relevant for Plansee, which is a clastic influenced lake and where macroscopic distinction between background sediment and a 1-2 mm flood layer is difficult. Hypothetically missing to delete 0.5cm of event layers equals 11 years (using overall mean sed.rate 0.045 cm/a), which is in most cases much smaller than the error of the age model. Although such an age shift would be higher in Piburgersee (0.5 cm = 20 years using overall mean sed.rate of 0.025 cm/a), it is still below the uncertainty range of the age model. Clastic event layers are much easier to detect in the organic-rich sediments of Piburgersee and thus, does not represent a problem at all.

Line 470: please provide the citations for the rockslides in the figure caption as you explicitly mention they are from previous work. They are best here when you first mention them rather than later on in the caption.

We addressed this comment now and citations are moved to the first appearance in Line 503

Minor issues with the supplement:

Line 18: could you please clarify this statement. Modified from what and how?

We addressed this by adding 'from a previous tectonic study' in Line 18

Supplementary figure 1: could you please change the fault line work to show the sense of motion i.e. thrust, strike slip etc.

We added more kinematic indicators to supplementary figure 1.

Line 85: attesting 'to' failure

We added the word 'the' in order to show that the observations indicate the failure of a local rock fall in Line 88.

Line 101: contrasting

We addressed this comment by changing the figure (Ca values are partly greyed out now) and further explained why geochemical are variable in the homogeneous deposit in supplementary lines 60 – 63.

Lines 152-155: the sentence is worded awkwardly making the meaning unclear. I think you are saying that brittle deformation overlain by liquidized deformation caused by a single earthquake is due to the contrasting physical properties of the sediment before deformation that includes lacustrine muds overlying a debrite.

Thank you for pointing out this unclear sentence. We followed the reviewer's suggestions and revised the sentence in order to provide more clarity in Lines 176-179.

Line 180: the 'exceeding' makes the '(or more)' redundant. Personally I would say earthquakes that produce seismic intensity of VI (or more)...

Given the fact that the CE 1930 Namlos earthquake had intensity VI at Piburgersee, but did not cause any specific sedimentary imprint, the earthquake recording threshold for Piburgersee is >VI.

Therefore, we have to keep the 'exceeding' but deleted the redundant phrase 'or more' in Line 209.

Line 203: in 'the deposit'.

Thanks for hinting at this mistake, which is now corrected in Line 251

Line 206: amplitude 'reflectors'

We added now 'reflection' amplitude in the sentence in Line 254, as we describe here the amplitudes of the reflections in seismic data and not the geological body (=reflector) causing the high amplitude.

Line 218-223: please reduce the complexity of this sentence to make reading it easier. The first part

of the sentence appears redundant.

We addressed this comment by deleting the first part of the sentence Line268-269.

REVIEWERS' COMMENTS

Reviewer #1 (Remarks to the Author):

Thank you for comprehensively addressing my review comments and for your revisions to the manuscript. I think the additional supplementary discussion and presentation of Supplementary Figure 15 has helped to provide balance in the consideration of different drivers of rock slope instability. This strengthens the argument of the role of earthquakes. With all of the other improvements too I am now satisfied that this manuscript is ready for publication; it provides an excellent contribution, well done! In preparing a final version, you may consider the following minor comments / suggestions:

Main manuscript

L30: I am not sure that 'could' is quite the correct tense for referring to the prehistoric rockslides. Perhaps this could instead be revised to "Some prehistoric rockslides were of extraordinary size, and if similar events were to occur today they would have devastating impacts. Currently our understanding of the causes of these prehistoric rockslides is hampered by...."

L72: Perhaps replace the word 'show' with 'present' or 'demonstrate' ?

Supplementary file

L188: should it be "We calibrated the intensity threshold REQUIRED to...."?

L365: I suggest you say here "Frost weathering (i.e. ice segregation and volumetric expansion) drive fracture propagation in a rock mass and, along with permafrost degradation, can cause progressive weakening of a rock slope" Otherwise you are ignoring the important ice segregation and permafrost degradation processes which are well established in the literature as drivers of slope instability.

I note that you mention here (L365) temperature-related drivers of instability (i.e. frost weathering), but you don't really mention temperature further in this section. The rest of the Supplementary Discussion 2 seems to focus on precipitation and ignores temperature. While I think this is probably fine (as I don't expect there to be a strong link to temperature changes either), given that you mention frost weathering you probably do need to address temperature for completeness. Can you add a sentence to this effect? I presume that the delta O-18 cave record may be presented as something of a temperature proxy (i.e. changes in glacier ice volume)??

Kind regards,
Sam McColl

Reviewer #2 (Remarks to the Author):

First of all congratulation and respect, you addressed most of my critical comments and points. I seldom found such a thorough and honest revised version. From my part I have no further comments, besides one. In the mean time Sybille Knapp published a paper on rock avalanches in the close by Eibsee (which is in your table of events), they provided new radiocarbon dating of c. 4 ka which fits your interpretation and event EQ2. These authors also regard local major seismic events as source. Please have a look: DOI 10.1002/esp.5025 Earth Surface Processes and Landforms. In my opinion this paper fosters your findings.

Reviewer #3 (Remarks to the Author):

To the editor and authors,

Thank you for the opportunity to review a revised version of the manuscript by Oswald et al. titled 'Seismic control of large prehistoric rockslides in the Eastern Alps'. I have carefully read the response to reviewers, the revised article and the supplement. My primary concern with the original submission was the absence of any discussion on the potential for hydro-climate to be both a trigger mechanism and a preparatory factor in the generation of rockfall clusters in the eastern European Alps. I suggested that to better support their argument that seismic shaking is a dominant trigger and preparatory factor in rockfall generation the authors should review and discuss Holocene paleoclimate archives to assess the potential of hydro-climate as a confounding variable. I am pleased to report that the authors have done a good job in collating relevant studies in the new Supplementary Figure 15 and discussed the implications of Holocene paleoclimate for their conclusions in the new Supplementary discussion 2.

The review of paleoclimate data shows that at least in the Eastern Alps there is no temporal association between hydroclimatic change involving higher precipitation or more frequent extreme precipitation events and the occurrence of the rockslide clusters. Hence, the authors infer that hydroclimate is an unlikely trigger mechanism in the case study they present. One caveat to this argument is the paucity of published paleoclimate archives from the eastern Alps compared to the central Alps. However, the lack of published records is mitigated to some degree by the fact that the authors use their own lake records to show hydrometeorological event frequency and sedimentation rate do not support hydroclimate as a dominant factor in rockfall generation. In my opinion the paleoclimate review, new figure and discussion improve the impact of the manuscript because its primary conclusion that earthquakes are an important trigger mechanism and preparatory factor in rockfall cluster generation is now better supported by the analysis presented. From my perspective the article is now suitable for publication in Nature Communications.

Kind regard,

Jamie Howarth

Minor comment:

Lines 369 of the supplement: remove 'beforehand' to increase readability because it is redundant.

REVIEWERS' COMMENTS

We respond to the reviewer comments in blue. Line numbers refer to the annotated manuscript and annotated supplementary information

Reviewer #1 (Remarks to the Author):

Thank you for comprehensively addressing my review comments and for your revisions to the manuscript. I think the additional supplementary discussion and presentation of Supplementary Figure 15 has helped to provide balance in the consideration of different drivers of rock slope instability. This strengthens the argument of the role of earthquakes. With all of the other improvements too I am now satisfied that this manuscript is ready for publication; it provides an excellent contribution, well done! In preparing a final version, you may consider the following minor comments / suggestions:

We thank the reviewer for this very positive comment and his thorough and detailed reviews especially on the sections about rockslides and their triggering and preparation mechanisms.

Main manuscript

L30: I am not sure that 'could' is quite the correct tense for referring to the prehistoric rockslides. Perhaps this could instead be revised to "Some prehistoric rockslides were of extraordinary size, and if similar events were to occur today they would have devastating impacts. Currently our understanding of the causes of these prehistoric rockslides is hampered by...."

We appreciate this suggestion of better phrasing these sentences, which we implemented in the new version in lines 29-31.

L72: Perhaps replace the word 'show' with 'present' or 'demonstrate' ?

We replaced it to "present" in Line 81.

Supplementary file

L188: should it be "We calibrated the intensity threshold REQUIRED to...."?

Thanks for pointing out this mistake. We added 'required' in order to be more clear in Line 184.

L365: I suggest you say here "Frost weathering (i.e. ice segregation and volumetric expansion) drive fracture propagation in a rock mass and, along with permafrost degradation, can cause progressive weakening of a rock slope" Otherwise you are ignoring the important ice segregation and permafrost degradation processes which are well established in the literature as drivers of slope instability. I note that you mention here (L365) temperature-related drivers of instability (i.e. frost weathering), but you don't really mention temperature further in this section. The rest of the Supplementary Discussion 2 seems to focus on precipitation and ignores temperature. While I think this is probably fine (as I don't expect there to be a strong link to temperature changes either), given that you mention frost weathering you probably do need to address temperature for completeness. Can you add a sentence to this effect? I presume that the delta O-18 cave record may be presented as something of a temperature proxy (i.e. changes in glacier ice volume)??

Thank you for the suggestion of the new sentence, which we implemented in the new version of the supplement in Line 356. We also added a small paragraph about frost weathering at the end of the supplementary discussion 2 in line 400-405.

Kind regards,
Sam McColl

Reviewer #2 (Remarks to the Author):

First of all congratulation and respect, you addressed most of my critical comments and points. I seldom found such a thorough and honest revised version. From my part I have no further comments, besides one. In the mean time Sybille Knapp published a paper on rock avalanches in the close by Eibsee (which is in your table of events), they provided new radiocarbon dating of c. 4 ka which fits your interpretation and event EQ2. These authors also regard local major seismic events as source. Please have a look: DOI 10.1002/esp.5025 Earth Surface Processes and Landforms. In my opinion this paper fosters your findings.

We thank the reviewer for this very positive comment and his detailed reviews especially on the paleoseismological part of the study. We have included the new age of Eibsee bei Knapp et al. 2020 and accordingly revised Figure 4, supplementary figure 15 and supplementary table 3. It is important to note that this new and more precise age of the Eibsee rockslides has no further implications for our results or interpretations, as also pointed out by the reviewer.

Reviewer #3 (Remarks to the Author):

To the editor and authors,

Thank you for the opportunity to review a revised version of the manuscript by Oswald et al. titled 'Seismic control of large prehistoric rockslides in the Eastern Alps'. I have carefully read the response to reviewers, the revised article and the supplement. My primary concern with the original submission was the absence of any discussion on the potential for hydro-climate to be both a trigger mechanism and a preparatory factor in the generation of rockfall clusters in the eastern European Alps. I suggested that to better support their argument that seismic shaking is a dominant trigger and preparatory factor in rockfall generation the authors should review and discuss Holocene paleoclimate archives to assess the potential of hydro-climate as a confounding variable. I am pleased to report that the authors have done a good job in collating relevant studies in the new Supplementary Figure 15 and discussed the implications of Holocene paleoclimate for their conclusions in the new Supplementary discussion 2.

The review of paleoclimate data shows that at least in the Eastern Alps there is no temporal association between hydroclimatic change involving higher precipitation or more frequent extreme precipitation events and the occurrence of the rockslide clusters. Hence, the authors infer that hydroclimate is an unlikely trigger mechanism in the case study they present. One caveat to this argument is the paucity of published paleoclimate archives from the eastern Alps compared to the central Alps. However, the lack of published records is mitigated to some degree by the fact that the authors use their own lake records to show hydrometeorological event frequency and sedimentation

rate do not support hydroclimate as a dominant factor in rockfall generation. In my opinion the paleoclimate review, new figure and discussion improve the impact of the manuscript because its primary conclusion that earthquakes are an important trigger mechanism and preparatory factor in rockfall cluster generation is now better supported by the analysis presented. From my perspective the article is now suitable for publication in Nature Communications.

We thank the reviewer for his detailed and thorough reviews especially on the limnogeological and rockslide-triggering sections.

Kind regard,

Jamie Howarth

Minor comment:

Lines 369 of the supplement: remove 'beforehand' to increase readability because it is redundant.

We deleted 'beforehand' in Line 363.